# Lack of tetrodotoxin analogues and individual metabolomic profiling of the cryptic frog *Colostethus imbricolus*

**Mabel Gonzalez**[1,2], **Pablo Palacios-Rodriguez**[3,4], **Chiara Carazzone**[1]*

**1** Department of Chemistry, Universidad de los Andes, Bogotá, Colombia, **2** Department of Biology, Stanford University, Palo Alto, California, United States of America, **3** Facultad de Estudios Ambientales y Rurales, Pontificia Universidad Javeriana, Bogotá, Colombia, **4** Department of Biological Sciences, Universidad de los Andes, Bogotá, Colombia

* c.carazzone@uniandes.edu.co

## Abstract

Poison frogs (Dendrobatoidea) are characterized by the great diversity of alkaloids discovered in their skin. However, most of these alkaloids have been found in brightly colored species and there is a wide lack of knowledge of alkaloid profiles in the less colorful species. Previous finding of paralytic tetrodotoxins (TTXs) in only two cryptically colored species from the genus *Colostethus,* establishes the unique occurrence of hydrophilic alkaloids in the superfamily Dendrobatoidea. Unpublished results using extracts from *Colostethus imbricolus*, demonstrated that this species contains paralysis-producing substances, after intraperitoneal injection of mice. To analyze their skin metabolites and to determine if they correspond to TTX, or TTX analogues, we have employed a TTX-targeted separation in normal phase gradient, and an untargeted profiling in reversed-phase gradient. After performing both analyses, neither TTX nor TTX-analogues were detected in *C. imbricolus*. In contrast, other metabolites were separated, allowing the extraction of 76 adducts common to both analyses, being 33 of them tentatively annotated as amphibian alkaloids, eight as amphibian metabolites different from alkaloids and 25 that matched with natural products from the DNP. A total of 10 common molecular formulas remained non-annotated. The absence of MS/MS spectra for these adducts requires their structures to be confirmed in future analyses, following the completion of targeted MS/MS acquisition. After analyzing the inter-individual variation of six specimens, it was demonstrated that the skin metabolome differs between males and females of *C. imbricolus*. Our results lead us to conclude that TTX is not the only paralyzing compound in dendrobatid frogs and that more work should be undergone to identify this phenomenon. A notable additional outcome of this study is the first successful separation of TTX on an SB-CN column using a normal-phase gradient, suggesting a potential useful approach for TTX-targeted separation.

**Data availability statement:** Most of relevant data are within the manuscript and its Supporting Information files. All converted files from both TTX targeted analysis in normal phase gradient and untargeted analysis in reversed-phase gradient generated during the current study are available in the the MassIVE online repository from the Global Natural Products Social Networking (GNPS) (MSV000086869-doi:10.25345/C5XR40 and MSV000086870-doi:10.25345/C5SZ22).

**Funding:** The research was supported by the announcement No. 757–2016 Doctorados Nacionales and project contract No. 44842–058-2018 from Ministerio Administrativo de Ciencia, Tecnología e Innovación (MINCIENCIAS). The financial support from the Faculty of Science at Universidad de los Andes partitioned in a forgivable loan assigned to one doctoral student (M.G.), the seed projects INV-2018-33-1259, INV-2019-67-1747 and FAPA project of C.C. The authors would like to thank the Vice Presidency of Research & Creation's Publication Fund and the Science Faculty at Universidad de los Andes for its financial support too. The funders had no role in study design, data collection and analysis, decision to publish, or preparation of the manuscript.

**Competing interests:** The authors declare that they have no competing interests.

## Introduction

Tetrodotoxin (TTX) is a potent neurotoxin that blocks sodium channels. It was first discovered on puffer fish or *fugu*, from where the name of the toxin is derived, for the order name Tetraodontiformes [1]. This toxin and their relative analogues are broadly distributed among different habitats (sea water, freshwater, land) and organisms, such as frogs, gastropods, crabs, newts, octopus, starfish, worms, copepods, bacteria, in addition to puffer fishes [2]. Dietary sources of TTX have been considered for marine organisms because non-toxic cultured puffer fish can acquire TTX through diet [3], and microorganisms are the potential original source because several compounds derived from TTX have been detected in marine bacteria [4]. For terrestrial organisms symbiotic TTX-producing bacteria in newts has been recently found [5], and previous evidence demonstrated that captive-reared species lack this toxin [6,7] but some amphibians are able to acquire TTX and metabolic biosynthetic intermediaries through dietary administration [8]. However, the biosynthetic pathway(s) of TTX and the organisms capable of synthesizing it remain unknown. Several studies suggest that a biosynthetic origin from a monoterpene precursor in amphibians cannot be ruled out [9–11]. Among amphibians, *Colostethus* genus if one of the genera were the hydrophilic TTX has been found but poorly studied. This genus belongs to the superfamily Dendrobatoidea. A group of poison frogs recognized because they have bright contrasting coloration as a visual warning signal to advertise their high toxicity [12]. Most of the toxic species attribute their toxicity to most of the 500 lipophilic alkaloids (not hydrophilic alkaloids) described [13,14]. *Colostethus panamensis* and *C. ucumari* are the exceptions. They are the only two species from the superfamily that contain hydrophilic alkaloids [15–17]. They also differ from their most recognized toxic relatives having dark cryptic coloration [18]. In fact, it is worth mentioning that for many years *Colostethus* was considered non-toxic. Then using behavioral mouse bioassays injecting frog skin extracts, the existence of water-soluble toxins was discovered [19]. It took 15 years using HPLC-FLD analysis to separate and identify tetrodotoxin (TTX), anhydroTTX and 4-epiTTX in *C. panamensis* [15], the first hydrophilic alkaloids found on the superfamiy Dendrobatoidea. No chemical analysis was developed with *C. ucumari*, but it was demonstrated that the aqueous extract from this species induced an alteration in mice behavior, which led to the hypothesis that TTX possibly was present [17].

Regarding the presence of lipophilic alkaloids, they have not been detected in any of the *Colostethus* species chemically surveyed. Additionally, dietary experiments feeding frogs with some lipophilic alkaloids indicate that the genus is unable to sequester these type of substances due to the lack of a metabolic mechanism designed for this purpose [20]. However, these experiments were carried out with only one species (*C. panamensis*) and with another species that is currently classified in another genus (*Allobates talamancae*), but which at that time was classified as *Colostethus*. The low number of species surveyed from this genus using more than a single skin for the analysis (2/19) [21] demonstrates how important it is to characterize the skin secretions from the other species in order to understand the evolution of water-soluble and lipophilic alkaloids as different anti-depredatory strategies in dendrobatids.

For detecting TTX or TTX analogues in another anuran species, like *Atelopus chiriquiensis*, *A. zeteki* [22], *Brachycephalus ephippium* [23,24], *B. nodoterga*, *B. pernix* [25], *Polypedates sp.* [26], researchers have used different extraction methods and methods based on Liquid Chromatography. For completing extraction, methanol or ethanol acidified with acetic acid was employed as solvent. For chemical analysis, most of the times LC-FLD with C-18 (ODS) columns were employed [23–26], while for LC-MS analysis researchers used the same columns combined with ion pair reagents such as ammonium heptafluorobutyrate [23–25]. Thanks to these analyses, it was discovered that 11-oxoTTX, a TTX analogue four to five times more toxic than TTX, was broadly found in many frogs, but rarely in marine animals [2,24]. 6-epi TTX, found in newts and salamanders, but not in frogs, is other analogue not usually found in aquatic animals [2].

For detecting lipophilic alkaloids, extraction with methanol, combined with alkaloid fractionation and GC-MS analysis, has been the most broadly applied procedure [27–29]. Additionally, the only database available for identification of amphibian alkaloids was designed for samples run on this platform [13]. However, as GC-MS restricts the analysis to volatile and semi-volatile compounds, some researchers have applied LC-MS for the analysis of lipophilic alkaloids [29–36], especially relevant for compounds with high molecular weights. Currently, metabolite annotation for LC-MS involves the creation of a personal LC-MS library with the accurate masses of frog alkaloids from Daly et al. database [13] or from LC-MS databases as the commercial Dictionary of Natural Products (DNP v.27.2, http://dnp.chemnetbase.com). The lack of a specific LC–MS/MS based database of lipophilic alkaloids ionized by electrospray (ESI) makes tentative identification more challenging. Then, fragmentation pathways for each of the compounds detected need to be proposed, and tools such CFM-ID v3.0 (available at https://cfmid3.wishartlab.com/), CFM-ID v4.0 (https://cfmid.wishartlab.com/), which generates *in silico* MS/MS spectra for suspected compounds, or CANOPUS (https://bio.informatik.uni-jena.de/software/canopus/) that classifies metabolites in different chemical taxonomies, can be useful. A manual inspection of correlations between MS/MS spectra and mass spectra fragments from the GC-MS Daly database could be feasible to select the most likely amphibian alkaloid(s) when several candidates with the same mass are detected. Furthermore, the lack of centralized information about hydrophilic compounds detected in amphibians further hinders the accurate annotation of amphibian metabolomes.

*C. imbricolus*, an endemic species from Chocó, Colombia, is the closest relative to *C. panamensis*. Both species shared the presence of yellowish to orange axillary and inguinal femoral spots, and the recent observation of pharmacological paralysis effects on mice for about 20 minutes, after intraperitoneal injection of methanolic extracts from *C. imbricolus* (P. Palacios-Rodriguez, A. Batista, D.Mejía-Vargas, A. Amézquita, unpublished results) motivated this study (see S1 Video). This frog is distributed in the department of Chocó at an altitude between 200 and 300 m.a.s.l. and is described as a riparian species (see Fig 1 for details about frog's habitat). It has brown coloration with two incomplete dorsolateral lines of golden color and its belly is black with blue dots [37] (see Fig 1). Since it was discovered, there have not been many studies on its ecology and natural history, much less on its toxicity. The only report on its toxicity refers to marginal effects observed on a mouse bioassay with a skin extract from a single specimen, but in that moment neither the presence of water-soluble toxins or lipophilic alkaloids could be demonstrated [38]. Taking into account the extensive quantitative variation found on TTX quantities between specimens of *C. panamensis* the need to test the toxicity of *C. imbricolus* using more samples of individual specimens [17] emerges. Since hydrophilic alkaloids have been detected in the genus, but lipophilic ones are commonly found in the superfamily to which this species belongs, we needed to design a methodology that allowed detecting both types of metabolites.

The aim of this research is oriented to characterize the compounds found on the skin of *C. imbricolus* and to determine if they correspond to TTX, TTX analogues, lipophilic alkaloids, or new compounds, and to analyze their inter-individual variation. To accomplish this goal, we have employed LC-MS analysis with two different approaches: i) TTX targeted separation in normal phase gradient and ii) untargeted profiling in reversed-phase gradient. The first approach is focused on tracking TTX and TTX analogues in frogs' samples, while the second is focused on tracking lipophilic alkaloids and other lipophilic compounds. In addition, we have set the experimental conditions for TTX detection on a cyano column with stable bond technology (Zorbax SB-CN), never used before for TTX-analysis.

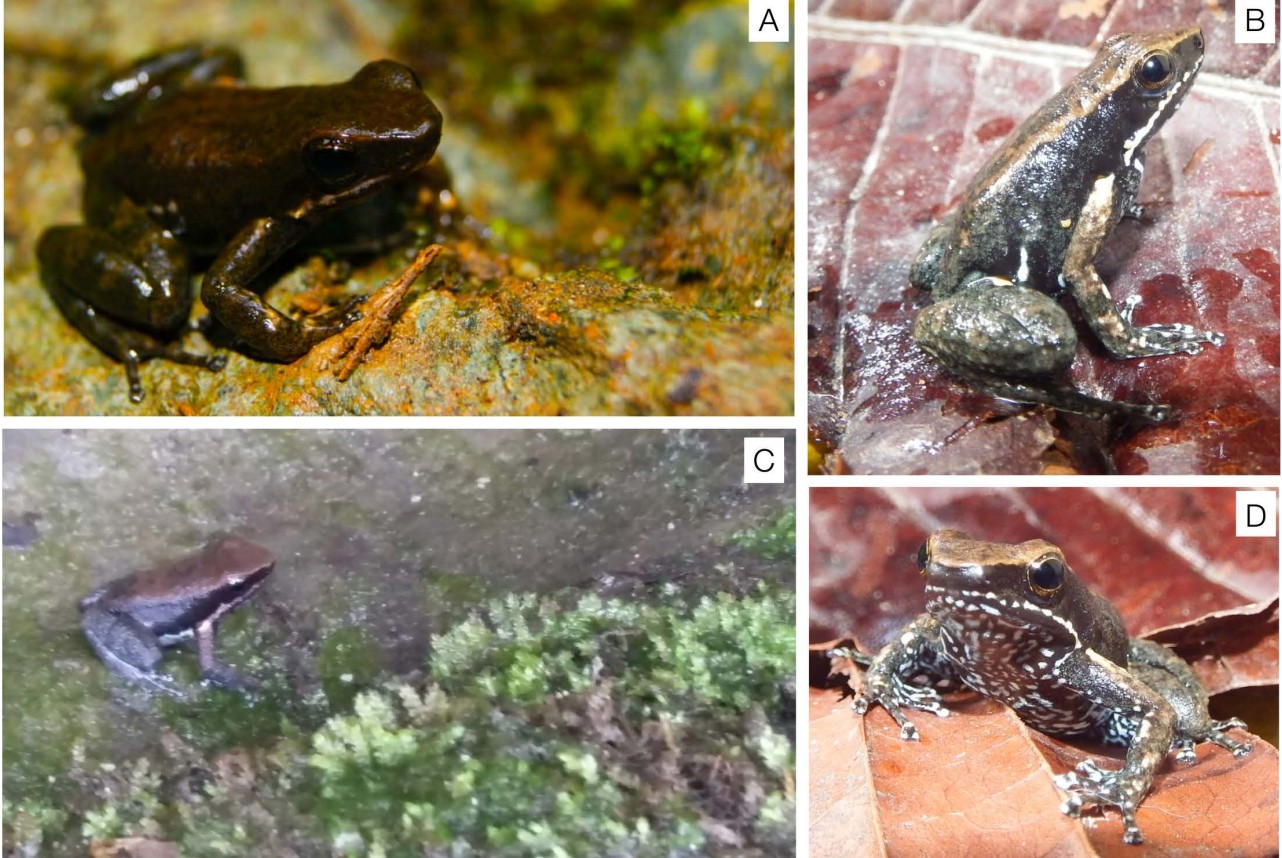

**Fig 1. Photographies of *Colostethus* species. A.** Photo of *C. panamensis*. **B.** Photo of *C. imbricolus*. **C.** Riparian habitat of *C. imbricolus*. **D.** Ventral color pattern of *C. imbricolus*. Photos reprinted from original images under a CC BY license, with permission from Alfredo Hernández-Díaz, Pablo Palacios-Rodríguez and Mabel Gonzalez, original copyright 2025.

## Materials and methods

### Chemicals

Methanol (MeOH) 98% was purchased from Honeywell (Michigan, USA), acetonitrile (ACN) from J.T. Baker (Palo Alto, CA, USA) and acetic acid (AA) from Sigma-Aldrich (USA). A standard of 1 mg of tetrodotoxin (TTX) was purchased from BocScience (New York, NY, USA). 5 g of trans-decahydroquinoline (DHQ), ammonium formate and formic acid were purchased from Sigma-Aldrich (USA). The ultrapure water was obtained from a water purification system Heal Force Smart-Mini (Shangai, China).

### Collection of animals

In October of 2016, 11 adult specimens of *C. imbricolus* were captured in the village Puerto Pervel, municipality of Cantón de San Pablo, Department of Chocó, Colombia. The habitat of this species is riparian where frogs usually perch on small stones or even inside water (Fig 1). A framework permit to conduct this study was provided by Autoridad Nacional de Licencias Ambientales (ANLA) in the resolution 1177 granted to La Universidad de los Andes for the Collection of Specimens of Wild Species of Biological Diversity for Non-Commercial Scientific Research Purposes. The animals were collected by visual encounter surveys and euthanized using medullary injection or "pithing". The methods used in this study

were carried out according to the regulations specified by the Institutional Animal Care and Use Committee of the the Faculty of Science of Universidad de los Andes, (COR_C.FUA_16–016). Subsequently, the complete skin was removed from each animal and stored in glass vials with PTFE/F217 lined caps (Ohio Valley, Marietta, OH, USA) containing 1 mL of MeOH. In addition to the skin, other organs such as liver, muscle, stomach, intestine and eggs (in females) were removed for further chemical analysis, but those results are going to be published successively. As several organs were removed from the specimens, they were not deposited in the museum of Universidad de los Andes. At the field, vials were pre-served at −4°C and transported in small coolers to the Laboratory of Advanced Analytical Techniques in Natural Products (LATNAP), at Universidad de los Andes. Then, samples were kept at −80°C.

For the chemical analyses two different sample preparation and LC-conditions were used employing different columns, chromatographic conditions, and MS/MS settings. One targeted approach for TTX and TTX-analogues detection in a normal-phase gradient, the other an untargeted analysis in a reversed-phase gradient method. The MS/MS acquisition mode for the targeted analysis was focused on tracing specific ions associated with TTX and TTX-analogues and STX, while the untargeted analysis was tracing all molecular ions that could be detected with the reversed-phase gradient using an auto MS/MS acquisition mode.

### TTX targeted analysis in normal phase gradient

**Sample preparation.** Frogs' skin methanolic extracts from 4 frogs were evaporated in a speed vacuum concentrator at room temperature (LABCONCO, Kansas City, MO, USA). The skin was previously cut in small pieces using stainless steel scissors and tweezers. Three skins (frog 9, frog 10, frog 11) were combined in a single extract (for assuring detectability) and one separate specimen (frog 8) was separated in two halves (arms and legs) in individual vials for the preliminary analysis performed using the first gradient method. Then, as metabolites were not fully dissolved in MeOH, they were reconstituted in 1 mL of 1% v/v AA following Bane et al. procedure for TTX analysis [39]. First, extracts were homogenized in a Bio-Gen PRO200 r (Pro Scientific, Oxford, CT, USA) in the same solvent for 3 min. Then, vortex (Heidolph, Schwabach, Germany) was applied for 3 min and samples were heated in boiling water for 8 min. Vortexed again during 30 s and cooled in ice water for 5 min. The following steps were 2 min of sonication and 15 min of centrifugation (5500 rpm, Thermo Scientific, San Jose, CA, USA). Then, the decanted supernatant was separated in a glass tube. 2 mL of solvent were added to the solid residue, vortex was applied for 30 s and centrifuged (3000 rpm) for 15 min. Two supernatants were combined and evaporated in a speed vacuum concentrator (LABCONCO, Kansas City, MO, USA). Then, samples were reconstituted in 1 mL of MeOH/H$_2$O (80:20, v/v) acidified to 1% with AA v/v. For the final gradient method, both extracts from frog 8 (arms and legs) were combined in a single extract.

**LC-Conditions.** A TTX standard solution was prepared solubilizing 1 mg of TTX in 1 mL of MeOH. Then, the working solutions were diluted to 10 ppm (~31 µM) (see Table 1 for details about TTX structure).

As TTX and TTX analogues are hydrophilic alkaloids, reversed phase HPLC columns need an ion pairing reagent such as ammonium heptafluorobutyrate for their analysis. High concentrations of this reagent suppress significantly the intensity of the sample signal [25,40], so other researchers have tried normal phase chromatography and HILIC columns [2]. In place, we employed a cyanopropylsilane column Zorbax SB-CN (150 x 3.0 mm x 3.5 um, Agilent technologies, Santa Clara, USA) for optimizing the separation conditions.

**Table 1. TTX properties relevant for mass spectrometry analysis.**

| Chemical structure | CAS | Mass | Molecular formula | Precursor ion [M+H]$^+$ | Product ions |
|---|---|---|---|---|---|
|  | 4368-28-9 | 319.1016 | C$_{11}$H$_{17}$N$_3$O$_8$ | 320.1088 | 302, 284, 256 |

Product ions obtained from Bane et al. [2].

We used two gradient methods, one of 30 minutes for optimizing TTX retention, and another of 20 minutes shortening the re-equilibration time. For both cases, the mobile phases were A (5 mM formic acid and 5 mM ammonium formate in Milli-Q water) and B (ACN, 5 mM formic acid) at a flow rate of 0.2 mL/min and 45ºC. First gradient elution started at 80%B for 3 min, then decreased to 70%B at 4 min, then to 65%B at 4.5 min, 60%B at 6 min, and maintained for 4 min, thereafter it returned to starting conditions in 15 min and finally 5 min of re-equilibration time was applied. Total run time was 30 min. Second gradient started at 80%B, then decreased to 70%B at 3 min, then to 60%B at 4 min, 55%B at 4.5 min, then increased again to 60%B at 6 min, 70%B at 7 min, and thereafter returned to starting conditions in 13 min. Total run time was 20 min. MS spectra of both methods were acquired in positive ion mode in the range of 20–400 m/z and a scan rate of 6 scan/s.

10 µL of either the TTX 10 ppm solution or of samples of frogs' extracts were analyzed using an HPLC1260 system coupled to a Q-TOF 6520 (Agilent Technologies, Santa Clara, CA, USA). The Q-TOF mass spectrometer was operated in positive electrospray ionization (ESI) mode in separate runs, employing a full scan mode in the mass range from 20 to 400 m/z and a scan rate of 6.0 scan/s. The mass spectrometer source conditions consisted of gas heater of 350 °C, a nebulizer gas flow rate of 13 L/min, a pressure of 45 psi, a capillary voltage of 3500V, fragmentor at 120 V, skimmer voltage of 75 V and octupole RF of 750 V. A constant mass correction was performed during all analysis using an Agilent Standard mix (Agilent Technologies, Santa Clara, USA) with two reference masses: m/z 121.0509 ($C_5H_4N_4$) and m/z 922.0098 ($C_{18}H_{18}O_6N_3P_3F_{24}$).

For MS/MS analysis, targeted MS/MS mode was employed for the precursor ion 320.1088 m/z, accurate mass value of $[C_{11}H_{18}N_3O_8]^+$, correspondent to the molecular ion in positive mode of TTX (See Table 1 for TTX properties). A collision energy of 24 V was used. This experiment was set for both injections, TTX and frogs' extracts, to run with the 30-minute gradient. Then, for the 20-minute gradient we performed a target MS/MS analysis in the mass range between 20–323 m/z and a scan rate of 6.0 scan/s for 9 precursor ions from TTX analogues and the precursor ion m/z 178.1342, the most intense molecular feature obtained from the frogs' preliminary analysis. This ion matched with one of the prevalent fragments (m/z 178) from the fragmentation pathways of TTX (2-aminodihydroxyquinazoline, $C_8H_7N_3O_2$) and other TTXs analogues [41]. We additionally traced saxitoxin (STX), another hydrophilic toxin, following preliminary analyses showing that metabolites of *C. imbricolus* exhibit high hydrophilicity and improved solubility in aqueous media. Collision energies and parameters for performing MS/MS analysis of these 11 precursor ions is presented in Table 2.

## Untargeted analysis in reversed-phase gradient

**Sample preparation.** A DHQ standard solution was prepared dissolving 1 mg in 1 mL of MeOH. Then, from this solution the working solutions were diluted to 10 ppm.

Frogs' skins methanolic extracts from 6 frogs were evaporated in the speed vacuum concentrator at room temperature (LABCONCO, Kansas City, MO, USA). Previously, the skin was cut in small pieces using stainless steel scissors and tweezers. Then, as metabolites were not fully dissolved in MeOH, they were dissolved in 500 µL of MeOH/$H_2$O (80:20, v/v) acidified to 1% v/v with AA and vortex for 1 min. Sample preparation was performed employing ultrasonic assisted extraction (UAE) for 15 min and then insoluble components were disregarded centrifuging samples to 5500 rpm for 15 min (Thermo Scientific, San Jose, CA, USA). 100 uL of supernatant were combined with 20 uL of DHQ solution at 10 ppm in a 250 uL micro-vial (Agilent Technologies, Santa Clara, CA, USA) and inserted to the HPLC autosampler.

**LC-Conditions.** Analyses were performed in a reversed-phase HPLC Zorbax SB-C18 column (150 mm x 2.0 mm x 3.0 um, Agilent technologies, Santa Clara, USA) for optimizing the separation conditions. The mobile phases were A (0.1% v/v formic acid in Milli-Q water) and B (0.1% v/v formic acid in can) at a flow rate of 0.3 mL/min and 45ºC. Gradient elution started at 5%B for 2 min, then increased to 20%B at 4 min, then to 30%B at 15 min, 34%B at 25 min, 45%B at 30, and finally to 100%B at 45 min, maintained for 5 min and thereafter returned to starting conditions in 2 min. Finally, 13 min of re-equilibration time were applied giving a total run time of 65 min. MS spectra were acquired in positive ion mode over the range of 20–2000 m/z and at a scan rate of 6 scans/min.

**Table 2. Experimental conditions employed for the targeted MS/MS mode analysis for tracking TTX and TTX analogues in samples from frogs' extracts.**

| Target mass | TTX, TTX-analogues or fragments traced | Target charge | Target retention time (min) | Target δ retention time (min) | Target isolation width (amu) | Target collision energy (V) |
|---|---|---|---|---|---|---|
| 178.1342 | $C_8H_7N_3O_2$ | 1 | 5 | 3 | 1 | 30 |
| 254.0000 | 4,9-anhydro-5,6,11-trideoxy TTX<br>4,9-anhydro-8-epi-5,6,11-trideoxy TTX<br>4,4a-anhydro-5,6,11- trideoxy TTX | 1 | 5 | 5 | 1 | 30 |
| 270.0000 | 1-hydroxy-4,4a-anhydro-8- epi-5,6,11-trideoxy TTX | 1 | 5 | 5 | 1 | 30 |
| 272.0000 | 5,6,11-trideoxy TTX<br>8-epi-5,6,11-trideoxy TTX<br>4-epi-5,6,11-trideoxy TTX | 1 | 5 | 5 | 1 | 30 |
| 288.0000 | 1-hydroxy-8-epi-5,6,11- trideoxy TTX<br>6,11dideoxyTTX<br>8,11dideoxyTTX | 1 | 5 | 3 | 1 | 30 |
| 290.0000 | 11-norTTX-6(S)-ol<br>11-norTTX-6(R)-ol | 1 | 5 | 3 | 1 | 40 |
| 300.0000 | Saxitoxin (STX) | 1 | 5 | 5 | 1 | 20 |
| 302.0983 | 4,9-anhydro TTX<br>6-epi-4,9-anhydroTTX<br>AnhydroTTX | 1 | 4 | 3 | 1 | 24 |
| 304.0000 | 5-deoxyTTX<br>11-deoxyTTX<br>1-hydroxy-8-epi-5,11-dideoxyTTX | 1 | 5 | 3 | 1 | 30 |
| 320.1088 | TTX<br>4-epi-TTX<br>6-epi-TTX<br>Tetrodonic acid | 1 | 5 | 7 | 1 | 24 |
| 336.0000 | 11-oxo-TTX<br>TTX | 1 | 5 | 3 | 1 | 36 |

*Collision energies values were set from values optimized by Rodríguez et al. [41]*

10 µL of each frog extract were injected using the same HPLC-Q-TOF system (Agilent Technologies, Santa Clara, CA, USA) employed for the targeted analysis. A few differences in the settings were the mass range for the full scan between 20–2000 m/z, the nebulizer gas flow rate of 8 L/min, pressure of 10 psi, and fragmentor of 175 V. The mass correction was performed in the same way as for the targeted analysis. For MS/MS analysis, auto MS/MS acquisition mode was employed for the analysis of one of the samples employing a mass range between 100–3000 m/z, a scan rate of 6.0 scans/s, isolation width of 1.3 amu, setting collision energies with automated ramps and excluding the most predominant solvent signal (m/z 142.1600).

Data-dependent spectrum files were converted from raw to mzML format using MSConvert (https://proteowizard.sourceforge.io/download.html). All converted files from both TTX targeted analysis in normal phase gradient and untargeted analysis in reversed-phase gradient generated during the current study are available in the the MassIVE online repository from the Global Natural Products Social Networking (GNPS) (MSV000086869 and MSV000086870).

## Data analysis

The LC-MS data processing was conducted using MZmine 3 software [42]. For the targeted analysis, extracted ion chromatograms (XIC) were obtained filtering the molecular formula of the precursor ions ([M + H]⁺) corresponding to TTX (Table 1) to track the presence of this alkaloid in three frog's extracts (FrogsMix, SK8-Legs and SK8-Arms). XIC

visualization and further comparison with a solution 10 ppm of TTX standard was performed. In case of TTX presence in one of the frogs' samples, an overlap peak should be observed between frogs and TTX standard. This peak should elute at the same retention time and the same product ions for the MS/MS analysis should be observed. XIC were exported in.pdf format and used to create Fig 2 on the software Graphic (https://apps.apple.com/us/app/graphic/id404705039?mt=12). Then, using MassHunter Qualitative Analysis (Agilent Software B06.00, Santa Clara, CA, USA) the same comparative analysis was also employed for tracking precursor ions from eight different TTX analogues (m/z 254, 270, 272, 288, 300, 302, 304, and 336) and for the most intense molecular feature observed in the preliminary targeted analysis (m/z 178.1342) (see Table 2). These results are summarized in the S1 File.

Targeted and untargeted analyses were completed building chromatograms, and completing feature detection filtering MS1 and MS2 with noise level above 3.0E3, and noise thresholds of 5.0 and 2.5, respectively. Retention times between 0.5 and 34 min were included on the LC–MS analysis, retention time tolerance was set to 0.4 min and m/z tolerances were set to 0.0050 or 10 ppm. Savitsky-Golay algorithm was applied to complete chromatogram smoothing. Base peak chromatograms (BPC) from the targeted analyses were extracted in MZmine to compare two chromatographic methods used and exported to.pdf format to build Fig 3. In contrast, BPC were extracted from each sample included in the untargeted analysis to visually compare chromatographic profiles obtained discriminating samples by sex (females and males). Then, using peak areas a Principal Component Analysis was completed coloring samples according to sex (females and males) and autoscaling based in SD and 1/5 of minimum missing value imputation. All plots obtained in MZmine for BPC and PCA were exported to.pdf format to build Figs 3 and 4 on the software Graphic.

A user database with amphibian alkaloids was created using the software Mass Hunter Qualitative analysis B.06.00. First, all molecular formulas of each amphibian alkaloid in Daly et al (2005) database and further publication were tabulated on an excel document. Then, the database was created using as a reference the PCDL database (Personal Compound Database and Library) and the default file for formula-database-generator. Our personal amphibian alkaloid database for MZmine is a.csv file that contains names, molecular formulas, and m/z of precursor ions of 912 metabolites extracted from this excel document (S2 Table). Annotation of targeted and untargeted analyses was completed tracking for positive and negative adducts ($H^+$, $Na^+$, $K^+$, $NH_4^+$, $H-H_2O+$, $H+2Na^+$, $2H^{2+}$, $M-H^-$, $-2H+Na^-$, $Cl^-$, $FA^-$) from our personal database. In addition, prediction of all molecular formulas exclusively including C, H, N and O was conducted using $[M+H]^+$ as a precursor ion. Corresponding aligned feature tables were exported to.cvs format. Molecular formulas predicted from sample SK8 from the targeted analysis were summarized in the S3 Table, while predicted molecular formulas candidates from the untargeted analysis in samples SK1-SK6 were summarized in the S4 Table. Final annotation of those molecular formulas detected simultaneously on both approaches (targeted and untargeted) was achieved with our personal amphibian alkaloid database (S2 Table). Then, remaining molecular formulas were compared with the database from the Dictionary of Natural Products (DNP v.27.2, http://dnp.chemnetbase.com) and literature with amphibian metabolites other than alkaloids [43–46].

## Results

### TTX targeted analysis in normal phase gradient

To illustrate TTX screening in *C. imbricolus,* two extracts were compared with the analytical standard of 10 ppm TTX and blank with MeOH. One of the frogs' extracts combined the skin of three frogs (frog 9, frog 10, frog 11), while the other contains only half of the skin from one specimen (arms of frog 8). Our results demonstrated that the corresponding precursor ion (320.1088) of TTX was not detected in these four specimens of *C. imbricolus* (Fig 2). The eluted peak of TTX at the retention time of 3.346 min (Fig 2D) was absent in the XIC from all frogs' samples (Fig 2B and 2C). Instead, XIC from frogs were indistinguishable from noise as it happens in a solvent blank (Fig 2A, 2B and 2C). MS/MS experimental fragmentation pattern from 10 ppm TTX solution (Fig 2E) were concordant with previous mass spectra obtained on ESI [40].

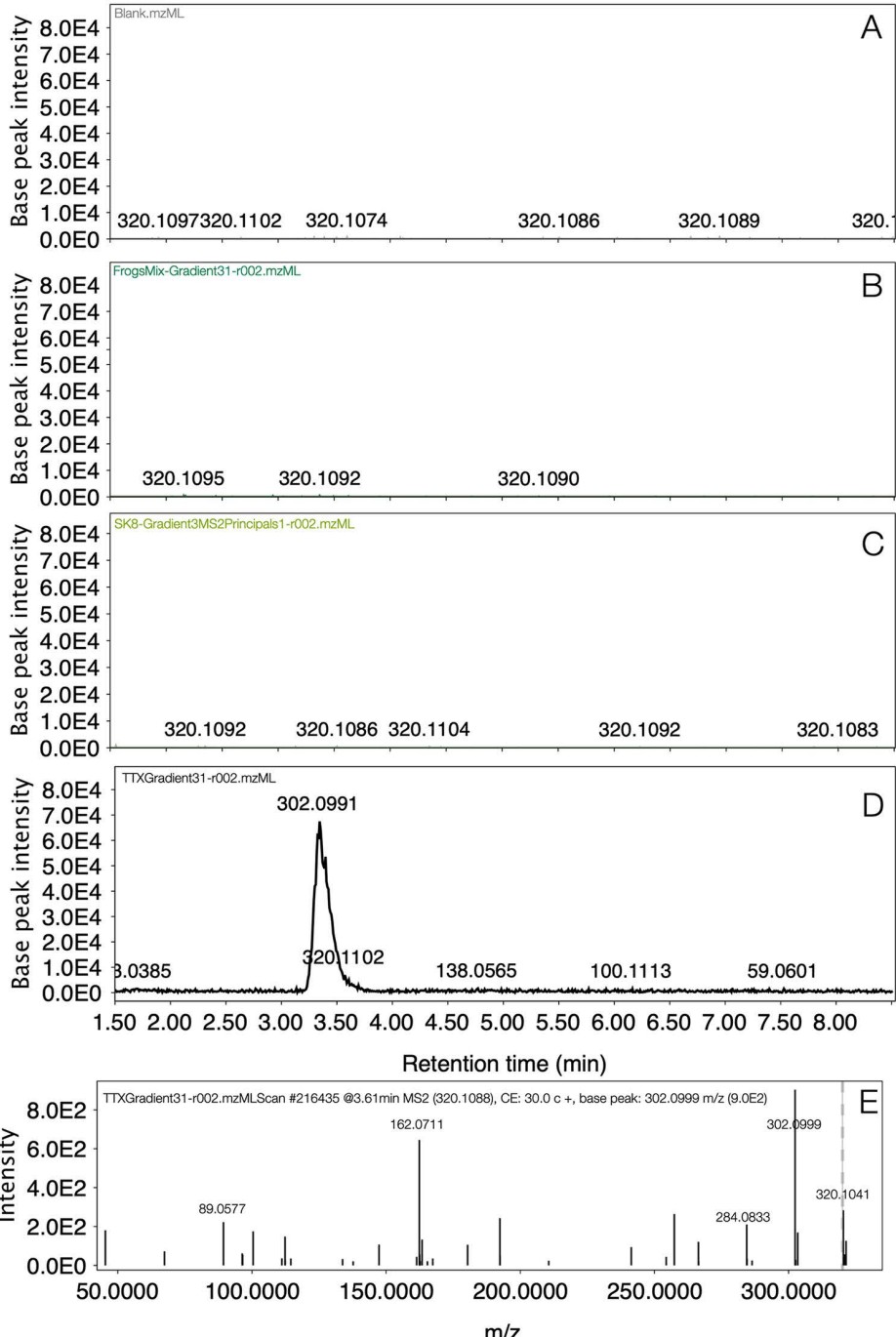

**Fig 2. TTX screening on *C. imbricolus* using a gradient method of 30 minutes in SB-CN column. A.** Base peak chromatograms (BPC) from target mass of TTX (320.1088) in MeOH blank. **B.** BPC from target mass of TTX (320.1088) in FrogsMix (frog 9, frog 10, frog 11). **C.** BPC from target mass of TTX (320.1088) in SK8 (arms of frog 8). **D.** BPC from target mass of TTX (320.1088) in solution of TTX 10 ppm. **E.** MS/MS from 10 ppm TTX standard.

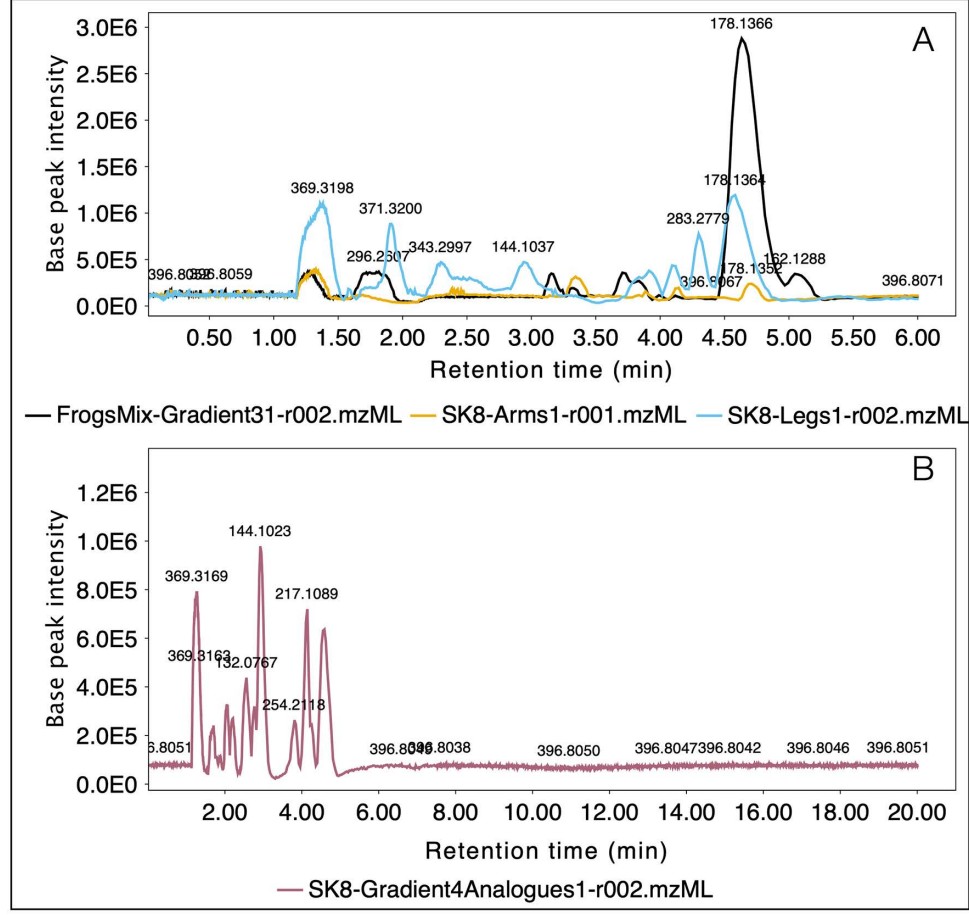

**Fig 3. Comparison of chromatographic profiles from *C. imbricolus* samples employing different gradient methods with SB-CN column. A.** BPC obtained from three samples employing a chromatographic method of 30 minutes. **B.** BPC obtained from one sample employing a chromatographic method of 20 minutes.

In contrast, using the same chromatographic method the BPC of the full-MS analysis from three extracts of *C. imbricolus* demonstrated that metabolites other than TTX could be extracted and analyzed. Visual comparisons between the chromatographic profiles revealed that all detected metabolites were eluted within the first 6 min (Fig 3A). Closer inspection shows that the sample containing skins from frogs 9 + 10 + 11 (FrogsMix) exhibited higher relative intensities of those metabolites, followed by the extract from the legs of frog 8 and then the arms' extract. Further comparison between skin sections from arms vs. legs from frog 8 also shows that some metabolites appear to be spatially distributed in a non-uniform way.

After verifying that the complete metabolome extracted from *C. imbricolus* eluted in about 6 minutes, we proceed to perform a modification on the preliminary gradient reducing their time to 20 minutes. The resulting profile of the combined extracts from arms and legs of frog 8 was very similar to those obtained with the 30-minute gradient for separated extracts, with a slight modification on retention times (Fig 3B). From this analysis 471 molecular features could be extracted, 424 were deconvoluted using molecular formula prediction from MZmine, and 94 of them matched with amphibian alkaloids (S3 Table).

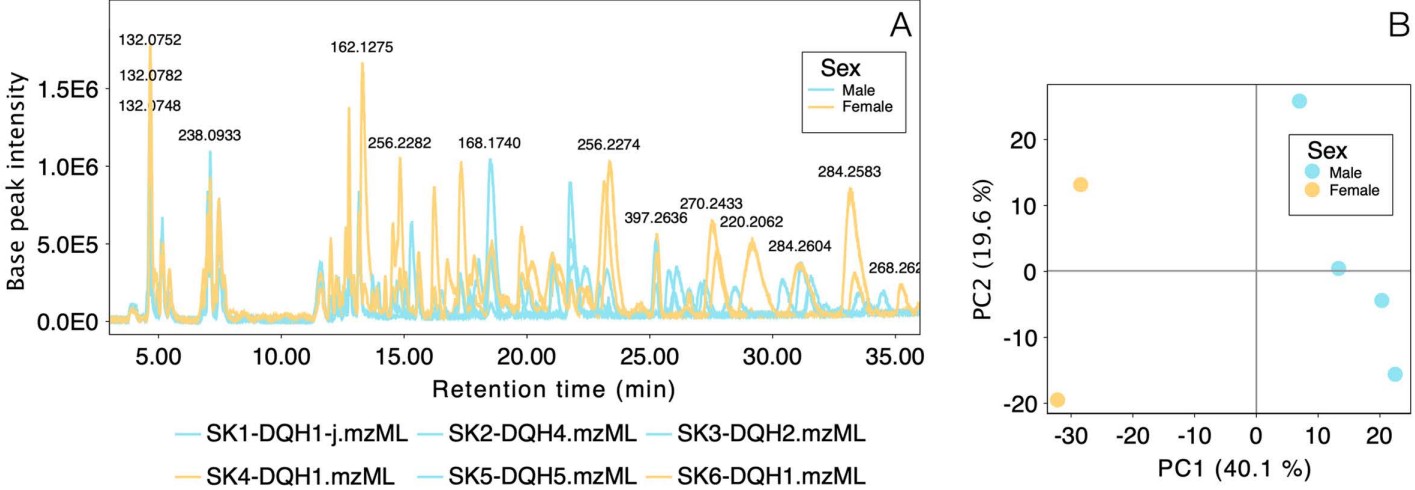

**Fig 4. Comparison of chromatographic profiles from six frog skins of *C. imbricolus* employing the SB-C18 column for untargeted analysis separating results by sex. A.** BPC obtained from two females and four males and masses of some molecular features. **B.** Principal component analysis (PCA) comparing peak areas of skin metabolites of females and males.

With this final gradient method, the screening of eight different target masses of TTX-analogues, STX and the precursor ion 178.1342 was performed (see Table 2 for experimental conditions of targeted MS/MS), but in contrast to TTX no analytical standards were used for comparisons. The BPC and corresponding MS/MS fragmentation patterns of these ions using Mass Hunter can be visualized in S1 File. While some nominal masses from Table 2 matched with metabolites detected in *C. imbricolus*, further inspection of their molecular formulas and MS/MS indicated that TTX, STX, and related TTX analogues were not detected in this species. From annotation results, m/z 254 was annotated as an amphibian alkaloid with molecular formula $C_{15}H_{27}NO_2$ that could correspond to allopumiliotoxin **243A**, and m/z 272 was annotated as an amphibian alkaloid with molecular formula $C_{14}H_{29}NO$ that could correspond to **277-Unclass** (Table 3). The final analysis of TTX on the 20-minute gradient allowed the elution of TTX at 3.175 min (a slight shift with the initial retention time of 3.346 min with the 30-minute gradient), and detailed MS and MS/MS analysis confirmed its identity (S1 File).

### Untargeted analysis in reverse-phase gradient

After confirming not detection of TTX, STX and TTX-analogues in *C. imbricolus*, individual metabolic profiling of six extracts (SK1 – SK 6) was completed in a SB-C18 column to gain a broader separation of the metabolites in a different chromatographic system. Comparison of the BPC from these specimens allowed the visualization of the inter-individual variation in their chemical profiles determined by sex (Fig 4A). Some prevalent peaks appeared in all specimens, while others varied. Between 0–36 min, the gradient method allowed us to resolve peaks very efficiently. Chemical differences among females and males were additionally supported by a PCA where PC1 is responsible of 40.1% of the explained variance and PC2 of 19.6% (Fig 4B). Using the top 10 loading values on extreme values of PC1 we can say that a lower value on PC1 is correlated with higher peak areas from features 1457, 2783, 3683, 2266 and 4125, as occurs in females. In contrast, males have higher values of PC1 correlated with higher peak areas from features 438, 391, 93, 51, and 436 (S5 Table).

From this analysis, a total of 1517 molecular features were extracted (S4 Table). To determine if the metabolites responsible for the paralysis effect in mice matched with molecular formulas of previously reported lipophilic amphibian alkaloids we used our personal amphibian alkaloid database for MZmine, finding that 440 different molecular features that

**Table 3. Tentative annotation of molecular features detected in *C. imbricolus* based on the comparison of their molecular formulas with our personal amphibian alkaloid database for MZmine, Dictionary of Natural Products (DNP) and additional amphibian literature.**

| Compound names | Molecular formula | Formula mass | Comp. adduct | SB-CN | | | SB-C18 | | |
|---|---|---|---|---|---|---|---|---|---|
| | | | | IDs that match | An. score | Δm ppm | IDs that match | An. score | Δm ppm |
| 153A-Unclass, 153B-Unclass, 153C-Unclass[a] | $C_{10}H_{19}N$ | 153.1518 | [M+H]+ | 340 | 0.930 | −1.82 | 1,845; 2,280; 2,405; 2,554; 2,699; 2,884 | 0.958 | −1.08 |
| Epilupinine and 19 additional hits[c] | $C_{10}H_{19}NO$ | 169.1467 | [M+H]+ | 309 | 0.792 | −2.080 | 649; 1,182; 1,548 | 0.991 | −0.09 |
| 196-Epiquinamide[a] | $C_{11}H_{20}N_2O$ | 240.1474 | [M+FA]– | 416 | 0.779 | −3.67 | 650; 985 | 0.756 | −4.05 |
| 167A-5,8-I, 167B Synthetic 5-propyl-Izidine, 167C-Unclass, 167D-Unclass, 167E-3,5-I, 167F-3,5-P, 167H-DHQ[a] | $C_{11}H_{21}N$ | 165.1491 | [M-H]– | 328 | 0.227 | −18.61 | 1,952 | 0.168 | −20.03 |
| 2-(2-Hydroxypropyl)-1-methyl-5-(2-oxopropyl)pyrrolidine and 5 additional hits[c] | $C_{11}H_{21}NO_2$ | 199.1572 | [M+H]+ | 296 | 0.956 | 0.440 | 1,643; 1,795 | 0.767 | 2.33 |
| N-Valylleucine and 8 additional hits[c] | $C_{11}H_{22}N_2O_3$ | 230.1630 | [M+H]+ | 233 | 0.909 | −0.910 | 1,577; 1,723; 2,102 | 0.966 | 0.34 |
| 185-Pip[a] | $C_{11}H_{23}NO$ | 167.1674 | [M+H-H2O]+ | 337 | 0.866 | −3.19 | 2,925; 3,109 | 0.946 | −1.28 |
| | $C_{11}H_{23}NO$ | 183.1623 | [M-H]– | 316; 217 | 0.693 | −6.66 | 967; 1,011; 1,525; 1,635; 1,797 | 0.642 | −7.78 |
| Unknown 1 | $C_{11}H_{49}N_2O_3$ | 257.3743 | [M+H]+ | 286 | 0.543 | 4.570 | 2,290; 2,570 | 0.794 | 2.06 |
| 3-Methylcarbonyloxyquinoline and 29 additional hits[c] | $C_{11}H_9NO_2$ | 187.0633 | [M+H]+ | 169 | 0.492 | 5.080 | 2,100 | 0.687 | 3.13 |
| 222/4A-N-Methylepibatidine[a] | $C_{12}H_{15}N_2Cl$ | 111.0433 | [M+2H]+2 | 141 | 0.301 | −24.95 | 296; 415; 750 | 0.356 | −23 |
| N,N-Dimethyltryptamine and 3 additional hits[c] | $C_{12}H_{16}N_2$ | 188.1314 | [M+H]+ | 407 | 0.714 | 2.860 | 2,513; 2,620; 2,662 | 0.672 | −3.28 |
| 191G-Unclass[a] | $C_{12}H_{17}NO$ | 191.1310 | [M+H]+ | 383 | 0.824 | 3.67 | 1,745 | 0.936 | −1.33 |
| | $C_{12}H_{17}NO$ | 208.1576 | [M+NH4]+ | 402 | 0.934 | 1.26 | 2,473 | 0.945 | 1.06 |
| 2-Hexyl-3,5-dimethylpyrazine (5 additional hits)[b] [43] | $C_{12}H_{20}N_2$ | 192.1627 | [M+H]+ | 386 | 0.886 | −1.140 | 2,827; 2,938; 3,142 | 0.947 | −0.53 |
| 197A-Unclass, 197C-5,8-I, 197D-Unclass, 197G-5,6,8-I, 197H-5,6,8-I, 197I-Izidine[a] | $C_{12}H_{23}NO$ | 197.1766 | [M+H]+ | 343 | 0.769 | −4.66 | 1,608 | 0.830 | −3.42 |
| 183A-Pip, 183B-Pyr, 183C-Unclass[a] | $C_{12}H_{25}N$ | 227.1885 | [M+FA]– | 332 | 0.787 | −3.73 | 1,760; 2,364; 2,442; 2,563; 2,704; 2,924 | 0.856 | −2.52 |
| 12-Aminododecanoic acid and 2 additional hits[c] | $C_{12}H_{25}NO_2$ | 215.1885 | [M+H]+ | 347 | 0.954 | −0.460 | 1,065; 1,184; 1,201 | 0.931 | −0.69 |
| 209A-DHQ, 209F-PTX, 209H-DesmethylhPTX, 209M-Unclass, 209O-Tricyclic, 209P-Tricyclic, 209Q-3,5-P, 209R-Unclass, 209S-5,8-I[a] | $C_{13}H_{23}NO$ | 209.1766 | [M+H]+ | 357 | 0.800 | −3.8 | 1,616 | 0.526 | −9.02 |
| 225E-aPTX, 225F-PTX[a] | $C_{13}H_{23}NO_2$ | 225.1729 | [M+H]+ | 331 | 0.986 | −0.25 | 2,261; 2,303 | 0.981 | −0.34 |
| 213A-Pip, 213B-Pip[a] | $C_{13}H_{27}NO$ | 235.1923 | [M+Na]+ | 333 | 0.705 | 4.99 | 1,992; 2,153 | 0.897 | 1.74 |
| 227-Unclass[a] | $C_{14}H_{29}NO$ | 271.2153 | [M+FA]– | 268 | 0.849 | −2.22 | 2,264 | 0.793 | −3.04 |
| Unknown 2 | $C_{14}H_{43}N_3$ | 253.3457 | [M+H]+ | 299 | 0.850 | −1.500 | 2,739; 3,070 | 0.552 | −4.48 |
| 251CC-Izidine[a] | $C_{15}H_{25}NO_2$ | 251.1885 | [M+H]+ | 324 | 0.949 | 0.81 | 2,189 | 0.960 | 0.63 |
| 297A-aPTX[a] | $C_{15}H_{27}N_4O$ | 279.2185 | [M+H-H2O]+ | 287 | | | 2,541; 2,560 | 0.677 | −4.62 |
| 237A-PTX[a] | $C_{15}H_{27}NO$ | 237.2093 | [M+H]+ | 398 | 0.897 | −1.72 | 2,461 | 0.944 | 0.94 |
| 253A-aPTX, 253C-Unclass, 253D-trans-DHQ, 253F-PTX, 253G-Tricyclic, 253M-Unclass, 253O-Unclass, 253Q-Unclass, 253S-Tricyclic[a] | $C_{15}H_{27}NO_2$ | 235.1936 | [M+H-H2O]+ | 397 | 0.984 | −0.27 | 2,360; 2,462; 2,716 | 0.964 | −0.61 |
| | $C_{15}H_{27}NO_2$ | 253.2042 | [M+H]+ | 298 | 0.950 | 0.79 | 3,207 | 0.992 | −0.12 |

*(Continued)*

| Compound names | Molecular formula | Formula mass | Comp. adduct | SB-CN | | | SB-C18 | | |
|---|---|---|---|---|---|---|---|---|---|
| | | | | IDs that match | An. score | Δm ppm | IDs that match | An. score | Δm ppm |
| 283B-Unclass, 283C-Unclass, 283D-Unclass, 283E-Unclass[a] | $C_{15}H_{29}N_4$ | 265.2392 | [M+H-H2O]+ | 387 | 0.690 | −4.66 | 4,083 | 0.662 | −5.08 |
| 239A-Izidine, 239AB-3,5-I, 239B-Izidine, 239C-5,8-I, 239CC-Izidine, 239 CD-3,5-I, 239D-5,8-I, 239DD-5,6,8-I, 239E-3,5-I, 239F-Izidine, 239G-5,8-I, 239H-HTX, 239K-cis-3,5-P, 239K-trans-3,5-P, 239L-Pip, 239Q-3,5-I, 239R-3,5-P, 239S-Isomer1-Unclass, 239S-Isomer2-Unclass, 239T-Unclass, 239U-5,8-I, 239W-5,6,8-I, 239X-Izidine, 239Y-3,5-P[a] | $C_{15}H_{29}NO$ | 239.2236 | [M+H]+ | 353; 203 | 0.921 | −1.31 | 3,715; 3,850; 3831 | 0.952 | 0.8 |
| 255A-Pip, 255C-Isomer1-Pip, 255C-Isomer2-Pip, 255D-Pip, 255E-Unclass[a] | $C_{15}H_{29}NO_2$ | 237.2084 | [M+H-H2O]+ | 352 | 0.879 | −2.03 | 1,991; 2,103; 2,391; 2,697 | 0.865 | −2.26 |
| | $C_{15}H_{29}NO_2$ | 255.2198 | [M+H]+ | 282 | 0.965 | 0.55 | 3,906 | 0.964 | −0.56 |
| 2,6-Bis(2-hydroxypentyl)piperidine and 13 additional hits[c] | $C_{15}H_{31}NO_2$ | 257.2355 | [M+H]+ | 285 | 0.882 | −1.180 | 2,660 | 0.997 | 0.03 |
| 267A-aPTX, 267B-Unclass, 267C-PTX, 267E-5,8-I, 267F-Unclass, 267G-Unclass, 267M-Unclass, 267N-DesmethylhPTX, 267P-hPTX, 267W-5,6,8-I[a] | $C_{16}H_{29}NO_2$ | 267.2203 | [M+H]+ | 307 | 0.919 | 1.21 | 3,247 | 0.919 | 1.21 |
| 253B-Isomer1–5,8-I, 253B-Isomer2–5,8-I, 253E-Unclass, 253H-5,6,8-I, 253K-5,6,8-I, 253L-Izidine, 253N-Unclass, 253P-Isomer1–5,6,8-I, 253P-Isomer2–5,6,8-I, 253R-Unclass, 253T-3,5-I, 253V-5,6,8-I[a] | $C_{16}H_{31}NO$ | 297.2309 | [M+FA]– | 278 | 0.876 | −1.66 | 2,318 | 0.864 | −1.83 |
| 241B-Unclass[a] | $C_{16}H_{35}N$ | 285.2668 | [M+FA]– | 289 | 0.756 | −3.41 | 3,026; 3,243; 3,315; 3,431 | 0.768 | −3.25 |
| 277B-PTX[a] | $C_{17}H_{27}NO_2$ | 277.2047 | [M+H]+ | 271; 392 | 0.932 | 0.98 | 3,031; 3,065; 3,098 | 0.887 | 1.62 |
| 279A-Unclass, 279J-Unclass, 279K-Unclass | $C_{17}H_{29}NO_2$ | 279.2198 | [M+H]+ | 354 | 0.939 | 0.870 | 3,655 | 0.973 | 0.38 |
| 281A-PTX, 281B-Isomer1-DeoxyPTX, 281B-Isomer2-DeoxyPTX, 281C-Unclass, 281D-Unclass, 281E-Unclass, 281F-DihydroPTX, 281G-Unclass, 281J-Unclass, 281K-hPTX, 281M-5,6,8-I, 281N-DeoxyPTX, 281O-5,8-I[a] | $C_{17}H_{31}NO_2$ | 263.2241 | [M+H-H2O]+ | 391 | 0.833 | −2.530 | 2,752; 2,865; 3,014 | 0.879 | −1.83 |
| | $C_{17}H_{31}NO_2$ | 281.2355 | [M+H]+ | 384; 391 | 0.992 | 0.120 | 2,624; 2,858; 3,185; 3,688; 3,749; 3,815; 2,752; 2,865; 3,014 | 0.985 | −0.21 |
| 283B-Unclass, 283C-Unclass, 283D-Unclass, 283E-Unclass[a] | $C_{17}H_{33}NO_2$ | 283.2516 | [M+H]+ | 288 | 0.886 | 1.600 | 2,971; 3,054; 3,249; 3,443; 3,574; 3,730 | 0.916 | 1.19 |
| 330-Pseudo[a] | $C_{18}H_{22}N_2O_4$ | 165.0776 | [M+2H]+2 | 256; 266 | 0.99 | −0.250 | 1285; 1,375 | 0.880 | −2.88 |
| Alanine (14 additional hits)[b] [44] | $C_3H_7NO_2$ | 89.0477 | [M+H]+ | 225 | 0.415 | 5.850 | 282 | 0.477 | −5.23 |
| Unknown 3 | $C_4H_5N_4O_4$ | 173.0311 | [M+H]+ | 148 | 0.818 | 1.820 | 437 | 0.858 | 1.42 |
| Creatinine[b] [44] | $C_4H_7N_3O$ | 113.0589 | [M+H]+ | 216 | 0.829 | 1.710 | 265 | 0.568 | −4.32 |
| Pyrrolidine[c] | $C_4H_9N$ | 71.0735 | [M+H]+ | 191 | −0.237 | 12.370 | 456 | 0.706 | 2.94 |
| dl-Alanine ethyl ester (49 additional hits)[b] [44] | $C_5H_{11}NO_2$ | 117.0790 | [M+H]+ | 186; 237; 292 | 0.992 | 0.080 | 685; 726 | 0.807 | 1.93 |
| Hypoxanthine (11 additional hits)[b] [44] | $C_5H_4N_4O$ | 136.0385 | [M+H]+ | 138 | 0.681 | 3.190 | 1,105 | 0.948 | −0.52 |
| 135-Adenine[a] | $C_5H_5N_5$ | 135.0545 | [M+H]+ | 159 | 0.837 | 4.79 | 774; 1,012; 1,698 | 0.905 | 2.78 |

*(Continued)*

**Table 3.** (Continued)

| Compound names | Molecular formula | Formula mass | Comp. adduct | SB-CN | | | SB-C18 | | |
|---|---|---|---|---|---|---|---|---|---|
| | | | | IDs that match | An. score | Δm ppm | IDs that match | An. score | Δm ppm |
| Isoguanine and 25 additional hits[c] | $C_5H_5N_5O$ | 151.0494 | [M+H]+ | 142 | 0.778 | 2.220 | 664; 696; 712; 730; 736; 925; 1,008; 1,081 | 0.992 | −0.08 |
| 2-Amino-6,8-dihydroxypurine and 6 additional hits[c] | $C_5H_5N_5O_2$ | 167.0443 | [M+H]+ | 69 | 0.819 | 1.810 | 805 | 0.887 | 1.13 |
| 5-Oxo-2-pyrrolidinecarboxylic acid and 23 additional hits[c] | $C_5H_7NO_3$ | 129.0426 | [M+H]+ | 196 | 0.626 | 3.740 | 663; 683; 709 | 0.622 | 3.78 |
| 2-Amino-3-methylbutanedioic acid and 50 additional hits[c] | $C_5H_9NO_4$ | 147.0532 | [M+H]+ | 164 | 0.683 | 3.170 | 325 | 0.605 | −3.95 |
| 5-Methyl-2-pyrrolidinecarboxylic acid and 90 additional hits[c] | $C_6H_{11}NO_2$ | 129.0790 | [M+H]+ | 226 | 0.934 | −0.660 | 146 | 0.941 | −0.59 |
| 4-Hydroxy-5-hydroxymethyl-2-pyrrolidinecarboxylic acid and 59 additional hits[c] | $C_6H_{11}NO_4$ | 161.0688 | [M+H]+ | 34; 168 | 0.924 | 0.760 | 424 | 0.716 | 2.84 |
| Ethyl methylalaninate and 75 additional hits[c] | $C_6H_{13}NO_2$ | 131.0946 | [M+H]+ | 173; 308; 419 | 0.810 | 1.900 | 670; 982; 1,046 | 0.760 | −2.4 |
| Arginine and 5 additional hits[c] | $C_6H_{14}N_4O_2$ | 174.1117 | [M+H]+ | 421 | 0.966 | 0.340 | 123; 127 | 0.797 | 2.03 |
| 3-Pyridinecarboxamide (2 additional hits)[b] [44] | $C_6H_6N_2O$ | 122.0480 | [M+H]+ | 146 | 0.855 | 1.450 | 391; 958; 1,021; 1,036 | 0.971 | −0.29 |
| 2-Amino-3-methylenehexanoic acid and 70 additional hits[c] | $C_7H_{13}NO_2$ | 143.0946 | [M+H]+ | 250 | 0.856 | 1.440 | 439 | 0.899 | 1.01 |
| 1-Acetyl-4-methylpiperazine and 1 additional hit[c] | $C_7H_{14}N_2O$ | 142.1106 | [M+H]+ | 432 | 0.654 | −3.460 | 576; 744; 1,352 | 0.929 | 0.71 |
| Theanine and 8 additional hits[c] | $C_7H_{14}N_2O_3$ | 174.1004 | [M+H]+ | 222 | 0.901 | 0.990 | 408 | 0.467 | 5.33 |
| Unknown 4 | $C_7H_{37}NO_2$ | 167.2824 | [M+H]+ | 338 | 0.955 | −0.450 | 2,927; 3,112 | 0.908 | 0.92 |
| Unknown 5 | $C_8H_{11}N_8$ | 219.1107 | [M+H]+ | 44 | 0.787 | −2.130 | 1,437 | 0.963 | −0.37 |
| Solsodomine B[c] | $C_8H_{13}N_3O$ | 167.1059 | [M+H]+ | 355 | 0.908 | −0.920 | 399 | 0.700 | −3 |
| Supinidine and 25 additional hits[c] | $C_8H_{13}NO$ | 139.0997 | [M+H]+ | 361 | 0.999 | 0.010 | 334 | 0.657 | −3.43 |
| N2-acetyllysine and 2 additional hits[c] | $C_8H_{16}N_2O_3$ | 188.1161 | [M+H]+ | 221; 245 | 0.919 | 0.810 | 1,472 | 0.668 | −3.32 |
| Unknown 6 | $C_8H_{17}N_4$ | 169.1453 | [M+H]+ | 388 | 0.882 | 1.180 | 961 | 0.711 | −2.89 |
| Sepiapterin (10 additional hits)[b] [45] | $C_9H_{11}N_5O_3$ | 237.0862 | [M+H]+ | 90 | 0.991 | −0.090 | 689 | 0.961 | −0.39 |
| Ichthyopterin (10 additional hits)[b] [45,46] | $C_9H_{11}N_5O_4$ | 253.0811 | [M+H]+ | 50 | 0.923 | 0.770 | 1,098 | 0.918 | −0.82 |
| Acetylcarnitine and 31 additional hits[c] | $C_9H_{17}NO_4$ | 203.1158 | [M+H]+ | 379 | 0.950 | −0.500 | 751; 2,880 | 0.884 | 1.16 |
| 2,2,6,6-Tetramethyl-4-piperidone oxime[c] | $C_9H_{18}N_2O$ | 170.1419 | [M+H]+ | 415 | 0.918 | −0.820 | 3,990 | 0.854 | −1.46 |
| 6-N-Trimethyllysine betaine and 6 additional hits[c] | $C_9H_{20}N_2O_2$ | 188.1525 | [M+H]+ | 450 | 0.975 | 0.250 | 192 | 0.868 | −1.32 |
| Unknown 7 | $C_9H_{55}N_3O_5$ | 285.4142 | [M+H]+ | 290 | 0.933 | 0.670 | 3,030 | 0.978 | 0.22 |
| Unknown 8 | $CH_{27}N_2O_4$ | 131.1971 | [M+H]+ | 178 | 0.511 | 4.890 | 634; 918; 986; 1,348 | 0.848 | 1.52 |
| Unknown 9 | $H_{10}N_5O_7$ | 192.0580 | [M+H]+ | 153 | 0.951 | 0.490 | 907 | 0.904 | −0.96 |
| Unknown 10 | $H_2O_{14}$ | 225.9445 | [M+H]+ | 275 | 0.938 | 0.620 | 163 | 0.582 | −4.18 |

[a] =Annotation completed using the personal database of amphibian alkaloids (S2 Table).

[b] =Annotations completed using the Dictionary of Natural Products (DNP) and literature with amphibian metabolites.

[c] =Annotations completed exclusively using the Dictionary of Natural Products (DNP).

For each chromatographic system we have detailed the molecular features deconvoluted with the corresponding adduct and molecular formula (column label as IDs that matched), the annotation score obtained in MZmine (column label as An. Score), and the lower mass difference from the list of features that matched (column label as Δm ppm).

matched with amphibian alkaloids (S2 Table). Additionally, four features strongly correlated with chemical differentiation between females and males in the PCA from Fig 4 (MF 2783: **223A**-5,6,8-I, 2266: **225A**-Izidine, 4125: **293G**-Unclass and 436: **135**-Adenine) were tentatively annotated as potential amphibian alkaloids while feature 391 was annotated as 3-pyridinecarboxamide (S5 Table). However, the absence of MS/MS fragmentation patterns for amphibian alkaloids makes it difficult to confirm this annotation.

## Comparison of metabolites extracted from targeted and untargeted analysis

Final manual comparison of those molecular formulas simultaneously detected on both approaches (targeted and untargeted) was achieved in excel, where tentative annotation and annotation scores of 76 adducts from 71 molecular formulas (Table 3) were detailed for each chromatographic system. First annotation of those molecular formulas was completed using our personal amphibian alkaloid database for MZmine (S2 Table) obtaining 33 matches with amphibian alkaloids, including potential pyrrolizidines (3,5-P), decahydroquinolines (DHQ), piperidines (Pip), epibatidines (Epi), indolizidines (3,5-I, 5,8-I, 5,6,8-I), izidines (Izidines), pumiliotoxins (PTX), allopumiliotoxins (aPTX), homopumiliotoxins (hPTX), tricyclic alkaloids (Tricyclic), histrionicotoxins (HTX) and unclass alkaloids (Unclass) (Table 3). Then, manual comparison of the molecular formulas of [M+H]+ adducts with the database from the Dictionary of Natural Products (DNP) and literature reporting amphibian metabolites other than alkaloids [43–46] (see Table 3) led to the annotation of eight compounds including 2-hexyl-3,5-dimethylpyrazine, alanine, creatinine, dl-alanine ethyl ester, hypoxanthine, 3-pyridinecarboxamide, sepiapterin and ichthyopterin. Next, 25 molecular formulas matched with one or several metabolites found in the DNP (but not amphibian literature). In this case we selected one of those matches for tentative annotation and detailed the total number of additional hits. Finally, 10 molecular formulas did not have any matches and remained non-annotated.

## Discussion

Chemical analysis performed on the cryptic dendrobatid *Colostethus imbricolus,* employing both normal-phase and reversed-phase chromatography, revealed that neither TTX or TXX-analogues were detected in this species (Fig 2 and S1 File). However, other metabolites different from these hydrophilic alkaloids have been separated employing a SB-CN column (for normal-phase HPLC), and SB-C18 column (for the reversed-phase approach). In spite, the enhanced solubility of frog's metabolites using a mixture of MeOH/H$_2$O (solvent highly hydrophilic), the best chromatographic separation was achieved using the reversed-phase gradient on the SB-C18 column, instead of the normal phase system (Fig 3A and B, versus Fig 4A).

 This finding supports the idea that *C. imbricolus* contains water-soluble compounds distinct from the TTXs reported in the closely related species *C. panamensis* [15]. In addition to this, the lack of TTXs in *C. imbricolus* also brings doubt about the type of compounds existent in *C. ucumari* and the possibility that results from mice-bioassays with this species could be attributed to TTXs [16,17] or other unknown water-soluble compounds. Our results agree with findings from Daly et al. in 1994 that also failed to detect hydrophilic alkaloids after analyzing an aqueous extract from a single skin of *C. imbricolus* [15]. Grant had suggested that maybe the lack of TTXs in these analyses could be attributed to the extensive quantitative variation found on TTX concentrations between specimens that some of them could be present, but undetectable at that moment [17]. Our results screening TTX presence in several specimens (frogs 9, 10 and 11) and sections (arms and legs sections from frog 8) employing a highly sensitive technique as targeted HPLC-ESI-qTOF undermine this hypothesis and provide convincing evidence that neither TTX and TTX-analogues are detected in *C. imbricolus*. Unfortunately, the lack of method validation for TTX analysis, as well as the absence of proper detection and quantification limit estimations for the SB-CN system, makes it difficult to determine whether TTX is truly absent in this species or simply undetectable. However, detecting ions with nominal masses that correspond to TTX-analogues (Table 2) but with fragmentation patterns that do not match previously published data [35–38], supports the idea that *C. imbricolus* lacks these compounds. The lack of MS/MS database for amphibian alkaloids makes it difficult to confirm our tentative annotations

of fragments m/z 254 and 272 as allopumiliotoxin **243A** and **277-Unclass** (Table 3), respectively. Molecular ion m/z 254 reported before by Daly in *C. ucumari* using GC-MRMS was deconvoluted as $[C_{10}H_{10}N_2S_3]^+$ and annotated as the artifact -bezothazolyl-N,N-dimethyl dithiocarbamate, originated from vial caps and septa [17]. However, LC-MS/MS information from our data (see fragmentation details from m/z 254 in the S1 File) are not relatable, suggesting that this molecular feature is unlikely to be the same artifact. Another important outcome from the targeted analysis, was reporting a chromatographic method for retaining TTX on a SB-CN column (150 x 3.0 mm x 3.5 um) employing a normal-phase gradient using water and ACN as mobile phases, both combined with 5 mM formic acid and 5 mM ammonium formate. Stable bond technology from Zorbax SB-CN column offered the necessary selectivity for TTX separation, and the advantage of being useful for normal-phase and reversed-phase separations [47]. This method is a new alternative, when TTX separation could not be performed in HILIC columns or when pair reagents are not suitable due to ion suppression effects.

Although using a reversed untargeted separation with the SB-C18 column provided better resolution (Fig 4A), highly polar compounds were not retained on this system and eluted with the solvent front. For example, the last peak eluted on the normal-phase gradient (Fig 3A y 3B), that corresponded to one of the most intense peaks in the chromatogram (m/z 178.1342, compound $C_{10}H_{15}N_3$), was not found in any specimen analyzed on the reversed-phase approach (Table 3). Likewise as mass range differed also between the targeted and untargeted analysis, the presence of other highly polar compounds like certain amphibian peptides [48], may have gone undetected due to the restricted mass range of 20–323 m/z used in the targeted analysis of TTXs.

In contrast, the reversed-phase untargeted analysis, with a mass range between 20–2000 m/z, provided a broader perspective of the chemical profile (detecting 1517 molecular features compared to 471 with the targeted approach). This method was used to compare the differences between females and males of *C. imbricolus.* PCA from this comparison, explained a total of 59.7% of variance and supported that sex determined chemical differentiation of the skin metabolome. This result aligns with other sex-based differences in this species, such as the coloration of orange/yellow spots on their bodies and parental care. Females are generally more conspicuous than males and number of transported tadpoles is directly correlated with the conspicuousness of the spots on females bodies [49]. As the function of skin metabolites is unknown, correlations between behavior and visual or chemical signals need to be studied in the future. For now it is interesting to see that some metabolites annotated as alkaloids (**223A**-5,6,8-I, **225A**-Izidine, **293G**-Unclass, **135**-Adenine and 3-pyridinecarboxamide) were associated with sex differentiation, while metabolites associated with pigmentation such as pterines (sepiapterin and ichthyopterin) [45,50], or the purine derivative hypoxanthine [46], have lower association with PC1 that supports sex differentiation (S5 Table). Inter-individual chemical differentiation in other dendrobatid species have been documented for specimens from different sex [51], populations [52,53], and developmental stages [54]. These variations in *C. imbricolus*, as in other dendrobatids, could be attributed to changes in abiotic factors, habitats, and availability of prey items [55].

A total of 471 molecular features have been detected in the analysis with the SB-CN column and 1517 molecular features in the analysis with the SB-C18 column. Detailed annotations of 76 adducts corresponding to 71 molecular formulas detected on both chromatographic systems was summarized in Table 3. Among these compounds 33 of them were annotated as an amphibian alkaloid. However, most of our tentative annotations remain unconfirmed, as the majority of these features lacked corresponding MS/MS spectra after the automatic MS/MS acquisition was completed. Despite our efforts performing MS/MS untargeted analysis, the results obtained on the qTOF instrument were not very informative. The annotation of m/z 254 (Table 3) illustrates the challenge of confirming an amphibian alkaloid from the Daly et al. (2005) database [13] using electron ionization (EI), even when LC-MS/MS fragmentation patterns are obtained via electrospray ionization (ESI) (S1 File). Until MS/MS fragmentation patterns for amphibian alkaloids are publicly available or more analytical standards are developed, limited comparisons between these fragmentation modes remain the primary resource for dendrobatid chemical analysis, but is not always possible. Discovery of lipophilic alkaloids in cryptic dendrobatids that have been presumed as non-chemically defended it has been documented before for *Silverstoneia punctriventris* [43] and

more recently in *Allobates insperatus, A. kingsburyi, A. talamancae, A. zaparo, Hyloxalus awa, Hyloxalus* sp. Agua Azul, *Leucostethus fugax, Rheobates palmatus, Silverstoneia* aff. *gutturalis, S. erasmios* and *S. flotator*, which suggests that presence of alkaloids may be more extensive than previously acknowledged [56].

Some of the other annotations (Table 3) corresponded to primary metabolites also found in the Siberian salamander (*Salamandrella keyserlingii*), including alanine and dl-alanine ethyl ester (from glycolysis), creatinine (originated from creatine degradation), hypoxanthine (originated from nucleotide degradation), and 3-pyridinecarboxamide (al. nicotinamide, originated from tryptophan metabolism) [44]. Additionally, annotation of the yellow pigment sepiapterin, previously reported in *Ranitomeya* species [45], and ichthyopterin, a blue fluorescent substance that have been previously isolated from fishes [57], demonstrates that LC-MS/MS allowed simultaneous detection of chemical defenses and pigments. Even one volatile organic compound (VOC) also reported in *S. punctriventris* [43], 2-hexyl-3,5-dimethylpyrazine, was also detected in *C. imbricolus*. The remaining annotations, based solely on the DNP were achieved by selecting plant alkaloids from the list of hits or very common primary metabolites. As such, they are equally probable as any other compound that matched the same molecular formula.

The quest for tracing the compounds responsible for altering mice-behavior in bioassays described on *C. imbricolus* endures and deserves further investigation. Future studies aimed to perform a bioactivity guided fractionation, and the corresponding MS/MS analysis from the metabolites detected are the first step for the identification of the paralytic metabolites found in the skin of *C. imbricolus*. Our combined use of targeted and untargeted approaches could also be applied to the chemical profiling of other dendrobatids in the future, enabling the simultaneous detection of hydrophilic (TTX/TXX-analogues) and lipophilic alkaloids.

## Conclusions

After completing an individual metabolomic profiling of the cryptic frog *Colostethus imbricolus,* using a TTX-targeted separation in normal phase gradient, and an untargeted profiling in reversed-phase gradient we concluded that neither tetrodotoxin or any tetrodotoxin analogue were detected in this species. The detection of 471 and 1517 molecular features, correspondingly on each chromatographic system, revealed how diverse is the range of polarities of the metabolites found on this frog. A total of 76 adducts were common to both analyses, and 33 of them were tentatively annotated as amphibian alkaloids. Using literature from amphibian metabolites and DNP, eight compounds were annotated as primary metabolites, pigments or VOC, while 25 matched with molecular formulas of plant metabolites found in the DNP. A total of 10 common molecular formulas remained non-annotated. The lack of fragmentation patterns of most of these features, and the lack of an amphibian alkaloid database obtained via electrospray ionization (ESI) demonstrates how challenging is annotating compounds of a non-model species as poison frogs, and how much work needs to be done using liquid chromatography to analyze chemical profiles of amphibians. Conducting this study, we also established a novel alternative for TTX separation in cases where HILIC columns or pair reagents are not suitable. Finally, individual profiling using the non-targeted approach allowed us to conclude that individual variation on the chemical profiles of this species is determined by sex. However, the ecological significance of this discovery, along with the identification of the compound responsible for inducing paralysis in mice is still under investigation.

## Supporting information

**S1 Video. Mice Bioassay video of *C. imbricolus* extract and saline solution control.**
(MP4)

**S1 File. Abstract (Spanish) and Supplementary information from chromatographic analysis.**
(DOCX)

**S2 Table. Personal amphibian alkaloid database for MZmine.**
(CSV)

**S3 Table. Molecular formulas predicted from targeted analysis.**
(CSV)

**S4 Table. Molecular formulas predicted from untargeted analysis.**
(CSV)

**S5 Table. Top 10 loading values from PC1 obtained from PCA.**
(CSV)

**S1 Fig. Graphical Abstract.**
(DOCX)

## Acknowledgments

We thank Leidy Alejandra Barragan-Contreras and to Adolfo Amézquita for their assistance in the field and to Lauren O'Connell for her advice creating the graphical abstract in BioRender.com.

## Author contributions

**Conceptualization:** Mabel Gonzalez, Chiara Carazzone.

**Data curation:** Mabel Gonzalez.

**Formal analysis:** Mabel Gonzalez.

**Funding acquisition:** Mabel Gonzalez, Chiara Carazzone.

**Investigation:** Mabel Gonzalez, Pablo Palacios-Rodriguez, Chiara Carazzone.

**Methodology:** Mabel Gonzalez, Pablo Palacios-Rodriguez, Chiara Carazzone.

**Project administration:** Chiara Carazzone.

**Resources:** Mabel Gonzalez, Pablo Palacios-Rodriguez, Chiara Carazzone.

**Supervision:** Chiara Carazzone.

**Visualization:** Mabel Gonzalez.

**Writing – original draft:** Mabel Gonzalez.

**Writing – review & editing:** Chiara Carazzone.

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
