## [Decision Letter · Decision Letter 0]

19 Sep 2025

Dear Dr. Carazzone,

Thank you for submitting your manuscript to PLOS ONE. After careful consideration, we feel that it has merit but does not fully meet PLOS ONE’s publication criteria as it currently stands. Therefore, we invite you to submit a revised version of the manuscript that addresses the points raised during the review process.

**ACADEMIC EDITOR:** 

Both reviewers saw merit in the manuscript, but reviewer one asks for additional methodological information (see below).

We look forward to receiving your revised manuscript.

Kind regards,

Renato Filogonio

Academic Editor

PLOS ONE

Journal Requirements:

[The research was supported by the announcement No. 757–2016 Doctorados Nacionales and project contract No. 44842–058-2018 from Ministerio Administrativo de Ciencia, Tecnología e Innovación (MINCIENCIAS). The financial support from the Faculty of Science at Universidad de los Andes partitioned in a forgivable loan assigned to one doctoral student (M.G.), the seed projects INV-2018-33-1259, INV-2019-67-1747 and FAPA project of C.C. The authors would like to thank the Vice Presidency of Research & Creation’s Publication Fund and the Science Faculty at Universidad de los Andes for its financial support too.].

3. Thank you for stating the following in your manuscript:

[The research was supported by the announcement No. 757–2016 Doctorados Nacionales and project contract No. 44842–058-2018 from Ministerio Administrativo de Ciencia, Tecnología e Innovación (MINCIENCIAS). The financial support from the Faculty of Science at Universidad de los Andes partitioned in a forgivable loan assigned to one doctoral student (M.G.), the seed projects INV-2018-33-1259, INV-2019-67-1747 and FAPA project of C.C. The authors would like to thank the Vice Presidency of Research & Creation’s Publication Fund and the Science Faculty at Universidad de los Andes for its financial support for the article processing charge.]

[The research was supported by the announcement No. 757–2016 Doctorados Nacionales and project contract No. 44842–058-2018 from Ministerio Administrativo de Ciencia, Tecnología e Innovación (MINCIENCIAS). The financial support from the Faculty of Science at Universidad de los Andes partitioned in a forgivable loan assigned to one doctoral student (M.G.), the seed projects INV-2018-33-1259, INV-2019-67-1747 and FAPA project of C.C. The authors would like to thank the Vice Presidency of Research & Creation’s Publication Fund and the Science Faculty at Universidad de los Andes for its financial support too.]

5. We note that Figure 1 in your submission may contain copyrighted images. All PLOS content is published under the Creative Commons Attribution License (CC BY 4.0), which means that the manuscript, images, and Supporting Information files will be freely available online, and any third party is permitted to access, download, copy, distribute, and use these materials in any way, even commercially, with proper attribution. For more information, see our copyright guidelines: http://journals.plos.org/plosone/s/licenses-and-copyright.

6. Please remove your figures from within your manuscript file, leaving only the individual TIFF/EPS image files, uploaded separately. These will be automatically included in the reviewers’ PDF.

Reviewers' comments:

Reviewer's Responses to Questions

**Comments to the Author**

1. Is the manuscript technically sound, and do the data support the conclusions?

Reviewer #1: Partly

Reviewer #2: Yes

2. Has the statistical analysis been performed appropriately and rigorously?

Reviewer #1: N/A

Reviewer #2: Yes

3. Have the authors made all data underlying the findings in their manuscript fully available?

Reviewer #1: Yes

Reviewer #2: Yes

4. Is the manuscript presented in an intelligible fashion and written in standard English?

Reviewer #1: Yes

Reviewer #2: Yes

Reviewer #1: The authors analyzed the hydrophilic toxins in the skin MeOH extract from the frog Colostethus imbricolus (11 specimens) collected in Colombia. Even though the frogs in this genus were previously reported to have paralytic toxins, they did not detected tetrodotoxin and its analogues using LCMSMS. I think that this is important finding. However, I could not find the detection limit and standard curves of TTX and its analogues for the analytical method they used in this manuscript. These descriptions are needed to be clearly shown.

Minor points

1. “Graphic abstract: Absence of TTX or any tetrodotoxin analogue”Should be revised to “Graphic abstract: Absence of TTX or any TTX analogues”

2. Table 2, why the molecular weights of 254.0000, 272.0000 and others are not the exactly accurate molecular weights of these analogues?

Reviewer #2: The manuscript titled “Lack of Tetrodotoxin analogues and individual metabolic profiling of the cryptic frog Colostethus imbricolus” (PONE-D-25-27319) includes a sophisticated series of analyses that allow the authors to investigate the presence of tetrodotoxin (TTX) and other alkaloid metabolites in a tropical dendrobatid frog species that has been proposed to possess TTX. The authors analyses indicate that TTX is not present in skin sample from the species but that a variety of other alkaloids are present. The authors use an interesting approach to catalog and begin to characterize these other natural products which will allow future investigation of this species as well as the natural products present in the species and genus.

The manuscript is well organized and complete. The experimental design of the project is solid, and I think the approach of using targeted and untargeted LC-MS/MS to investigate TTX and other alkaloids is an elegant solution to a tricky problem. The presence/absence of TTX in the genus Colosthethus is a long-term question that is of interest to ecologists, natural product chemists, and conservation biologists. My sense is that reporting negative results (e.g., the absence of TTX) is important and warrants publication. Because the manuscript also includes a more complete and interesting analysis of the metabolome of the species, I think it warrants publication in PLOS ONE.

The methods and results are clear, and the conclusions and discussion of the results are warranted. My training does not allow me to assess the details of their MS/MS analysis (Table 2) or the data analysis of the metabolomic profiling. However, it appears to me that the authors have provided adequate details for their approach to be replicated and/or judged by someone with more direct experience. Other elements of their analysis seem sound, and I am satisfied that their “null” results and absence of TTX in their samples is accurate and not an experimental artifact.

The discussion and conclusions in the manuscript seem warranted and appropriate for the data set. As with the MS/MS analyses by training does not allow me to assess the more granular level discussion of their data analyses but the broad conclusions and associated with the metabolic analyses appear to be accurate and supported by the data. This approach is a useful tool for building a more complete understanding of the chemical ecology of amphibians and was excited to see it included here.

A minor concern of the paper is the introduction and discussion of the origin of TTX. I am not sure that results presented in the manuscript are relevant to the ultimate origin of TTX as discussed in lines 57 to 62. There is still some disagreement about this. If the authors want to include this discussion, I encourage them to also include recent work from the Yotsu-Yamashita lab that suggests the possibility that TTX-bearing salamanders may synthesize their own TTX and at least acknowledge that for some amphibians the issue is still unresolved.

Overall, the manuscript warrants publication with some minor revisions and clean up. I have identified some specific concerns below:

Specific Concerns:

Lines 58-63. See above, this section may not belong in this paper at all but should include recent paper by Kudo from the Yotsu-Yamashita lab that maps a potential synthesis pathway of TTX from salamander skin intermediates that may not require bacteria.

Lines 70-73. I understand this sentence (I think), but the structure was confusing and could be clarified.

Line 79. This substance or these substances not as written.

Lines 91-93. Why are the authors only talking about 11-oxoTTX here? There are other analogs (e.g., 6-epiTTX) that also have terrestrial/marine distribution biases. Why discuss one but not the other?

Materials and Methods

I thought this section was well written and complete. However, one small request would be to include the molar concentration of their TTX standard in addition to the PPM concentration. It makes it easier for non-chemists to get a sense of detection limits with these approaches.

Another question that may represent ignorance on my part is the use of targeted and untargeted with reference to MS/MS analysis. As somebody slightly outside the field it took me a bit to realize what the terms were referencing (difference is MS/MS mode) it might help a broader audience to understand the results and design of the project to clarify why the two different approaches in the introduction.

Discussion

Lines 416-419. This sentence was confusing to me “in spite” of what? I was not sure what the first part of sentence was.

One thing that seemed missing from the discussion was explicit confirmation that none of the TTX analogs, potential precursors, and/or TTX metabolites (e.g., anhyrdro-TTX) are present in the metabolomes. Kudo et al. “Structures of N-Hydroxy-Type Tetrodotoxin Analogues and Bicyclic Guanidinium Compounds Found in Toxic Newts” document a range of compounds using MS/MS detection. Are these compounds included on the comparative database or annotations? Can the author’s results be used to confirm that other TTX related compounds are not present in their samples? This may not be possible, but it would be of interest if it could be confirmed.

**Do you want your identity to be public for this peer review?** For information about this choice, including consent withdrawal, please see our For information about this choice, including consent withdrawal, please see our Privacy Policy .

Reviewer #1: No

Reviewer #2: No

While revising your submission, please upload your figure files to the Preflight Analysis and Conversion Engine (PACE) digital diagnostic tool, https://pacev2.apexcovantage.com/ . PACE helps ensure that figures meet PLOS requirements. To use PACE, you must first register as a user. Registration is free. Then, login and navigate to the UPLOAD tab, where you will find detailed instructions on how to use the tool. If you encounter any issues or have any questions when using PACE, please email PLOS at . PACE helps ensure that figures meet PLOS requirements. To use PACE, you must first register as a user. Registration is free. Then, login and navigate to the UPLOAD tab, where you will find detailed instructions on how to use the tool. If you encounter any issues or have any questions when using PACE, please email PLOS at figures@plos.org . Please note that Supporting Information files do not need this step.. Please note that Supporting Information files do not need this step.

---

## [Author Response · Author response to Decision Letter 1]

25 Oct 2025

We have added individual responses to Journal requirements and each reviewer comment from the decision letter in the document "Response to reviewers". We highly appreciate all your suggestions.

Journal Requirements:

Response. We have corrected the format for author affiliations.

[The research was supported by the announcement No. 757–2016 Doctorados Nacionales and project contract No. 44842–058-2018 from Ministerio Administrativo de Ciencia, Tecnología e Innovación (MINCIENCIAS). The financial support from the Faculty of Science at Universidad de los Andes partitioned in a forgivable loan assigned to one doctoral student (M.G.), the seed projects INV-2018-33-1259, INV-2019-67-1747 and FAPA project of C.C. The authors would like to thank the Vice Presidency of Research & Creation’s Publication Fund and the Science Faculty at Universidad de los Andes for its financial support too.].

Response. We have edited the Funding Statement following your instructions and included this text within the cover letter.

3. Thank you for stating the following in your manuscript:

[The research was supported by the announcement No. 757–2016 Doctorados Nacionales and project contract No. 44842–058-2018 from Ministerio Administrativo de Ciencia, Tecnología e Innovación (MINCIENCIAS). The financial support from the Faculty of Science at Universidad de los Andes partitioned in a forgivable loan assigned to one doctoral student (M.G.), the seed projects INV-2018-33-1259, INV-2019-67-1747 and FAPA project of C.C. The authors would like to thank the Vice Presidency of Research & Creation’s Publication Fund and the Science Faculty at Universidad de los Andes for its financial support for the article processing charge.]

[The research was supported by the announcement No. 757–2016 Doctorados Nacionales and project contract No. 44842–058-2018 from Ministerio Administrativo de Ciencia, Tecnología e Innovación (MINCIENCIAS). The financial support from the Faculty of Science at Universidad de los Andes partitioned in a forgivable loan assigned to one doctoral student (M.G.), the seed projects INV-2018-33-1259, INV-2019-67-1747 and FAPA project of C.C. The authors would like to thank the Vice Presidency of Research & Creation’s Publication Fund and the Science Faculty at Universidad de los Andes for its financial support too.]

Response. We have deleted the Funding Statement from the manuscript.

Response. We already had the information from the ethics statement included in the Methods section, so we deleted it from the Statements and Declarations section.

5. We note that Figure 1 in your submission may contain copyrighted images. All PLOS content is published under the Creative Commons Attribution License (CC BY 4.0), which means that the manuscript, images, and Supporting Information files will be freely available online, and any third party is permitted to access, download, copy, distribute, and use these materials in any way, even commercially, with proper attribution. For more information, see our copyright guidelines: http://journals.plos.org/plosone/s/licenses-and-copyright.

Response. We have added three proof of granted permissions to our submission and details about CC license to figure captions.

6. Please remove your figures from within your manuscript file, leaving only the individual TIFF/EPS image files, uploaded separately. These will be automatically included in the reviewers’ PDF.

Response. We have removed all figures files from the manuscript.

Reviewers' comments:

5. Review Comments to the Author

Reviewer #1: The authors analyzed the hydrophilic toxins in the skin MeOH extract from the frog Colostethus imbricolus (11 specimens) collected in Colombia. Even though the frogs in this genus were previously reported to have paralytic toxins, they did not detected tetrodotoxin and its analogues using LCMSMS. I think that this is important finding. However, I could not find the detection limit and standard curves of TTX and its analogues for the analytical method they used in this manuscript. These descriptions are needed to be clearly shown.

Response. Unfortunately we did not make a method validation for the analysis of TTX in either of the chromatographic methods presented, as the original aim of the study was to separate and identify the metabolites contained in the skin of C. imbricolus. Acknowledging this big limitation of our research we have changed “absence” for “not detected” in lines 448, 452, 487, 496 of the document with track changes. Additionally, in the discussion section we have added this statement in lines 582-586: “Unfortunately, the lack of method validation for TTX analysis, as well as the absence of proper detection and quantification limit estimations for the SB-CN system, makes it difficult to determine whether TTX is truly absent in this species or simply undetectable. However, detecting ions with nominal masses that correspond to TTX-analogues (Table 2) but with fragmentation patterns that do not match previously published data [35–38], supports the idea that C. imbricolus lacks these compounds.

Minor points

1. “Graphic abstract: Absence of TTX or any tetrodotoxin analogue”Should be revised to “Graphic abstract: Absence of TTX or any TTX analogues”

Response. We changed the graphical abstract to “No detection of TTX or any TTX analogues”

2. Table 2, why the molecular weights of 254.0000, 272.0000 and others are not the exactly accurate molecular weights of these analogues?

Response. We used the parameters used by Rodríguez et al (2017) in the Table 1 on their supplementary material as a reference to create our chromatographic method. In their study, nominal masses were used because the analyses were performed in MRM mode on a triple quadrupole mass spectrometer. When adapting their method to our Q-TOF mass spectrometer, we also used nominal masses with a ±1 u window instead of tracing accurate precursor ions. We acknowledge this limitation and apologize for it; however, this approach still allowed us to effectively adapt the method to our instrument and monitor the target compounds.

Reviewer #2: The manuscript titled “Lack of Tetrodotoxin analogues and individual metabolic profiling of the cryptic frog Colostethus imbricolus” (PONE-D-25-27319) includes a sophisticated series of analyses that allow the authors to investigate the presence of tetrodotoxin (TTX) and other alkaloid metabolites in a tropical dendrobatid frog species that has been proposed to possess TTX. The authors analyses indicate that TTX is not present in skin sample from the species but that a variety of other alkaloids are present. The authors use an interesting approach to catalog and begin to characterize these other natural products which will allow future investigation of this species as well as the natural products present in the species and genus.

The manuscript is well organized and complete. The experimental design of the project is solid, and I think the approach of using targeted and untargeted LC-MS/MS to investigate TTX and other alkaloids is an elegant solution to a tricky problem. The presence/absence of TTX in the genus Colosthethus is a long-term question that is of interest to ecologists, natural product chemists, and conservation biologists. My sense is that reporting negative results (e.g., the absence of TTX) is important and warrants publication. Because the manuscript also includes a more complete and interesting analysis of the metabolome of the species, I think it warrants publication in PLOS ONE.

The methods and results are clear, and the conclusions and discussion of the results are warranted. My training does not allow me to assess the details of their MS/MS analysis (Table 2) or the data analysis of the metabolomic profiling. However, it appears to me that the authors have provided adequate details for their approach to be replicated and/or judged by someone with more direct experience. Other elements of their analysis seem sound, and I am satisfied that their “null” results and absence of TTX in their samples is accurate and not an experimental artifact.

The discussion and conclusions in the manuscript seem warranted and appropriate for the data set. As with the MS/MS analyses by training does not allow me to assess the more granular level discussion of their data analyses but the broad conclusions and associated with the metabolic analyses appear to be accurate and supported by the data. This approach is a useful tool for building a more complete understanding of the chemical ecology of amphibians and was excited to see it included here.

A minor concern of the paper is the introduction and discussion of the origin of TTX. I am not sure that results presented in the manuscript are relevant to the ultimate origin of TTX as discussed in lines 57 to 62. There is still some disagreement about this. If the authors want to include this discussion, I encourage them to also include recent work from the Yotsu-Yamashita lab that suggests the possibility that TTX-bearing salamanders may synthesize their own TTX and at least acknowledge that for some amphibians the issue is still unresolved.

Overall, the manuscript warrants publication with some minor revisions and clean up. I have identified some specific concerns below:

Specific Concerns:

Lines 58-63. See above, this section may not belong in this paper at all but should include recent paper by Kudo from the Yotsu-Yamashita lab that maps a potential synthesis pathway of TTX from salamander skin intermediates that may not require bacteria.

Response. Thank you so much for this interesting suggestion, it is completely true and we added this fact to the introduction (lines 66-67 of the version with track changes).

Lines 70-73. I understand this sentence (I think), but the structure was confusing and could be clarified.

Response. We edited the text: “Then using behavioral mouse bioassays injecting frog skin extracts, the existence of water-soluble toxins was discovered [16]. It took 15 years using HPLC-FLD analysis to separate and identify tetrodotoxin (TTX), anhydroTTX and 4-epiTTX in C. panamensis [12], the first hydrophilic alkaloids found on the superfamiy Dendrobatoidea.”

Line 79. This substance or these substances not as written.

Response. Text edited to “these substances”.

Lines 91-93. Why are the authors only talking about 11-oxoTTX here? There are other analogs (e.g., 6-epiTTX) that also have terrestrial/marine distribution biases. Why discuss one but not the other?

Response. We were more focused on the chromatographic method for that sentence, but we added the major distribution of 6-epi TTX in terrestrial animals too.

Materials and Methods

I thought this section was well written and complete. However, one small request would be to include the molar concentration of their TTX standard in addition to the PPM concentration. It makes it easier for non-chemists to get a sense of detection limits with these approaches.

Response. We have added the equivalency of 10 ppm to ~31 uM

Another question that may represent ignorance on my part is the use of targeted and untargeted with reference to MS/MS analysis. As somebody slightly outside the field it took me a bit to realize what the terms were referencing (difference is MS/MS mode) it might help a broader audience to understand the results and design of the project to clarify why the two different approaches in the introduction.

Response. In lines 283-286 we have added a short definition of each approach. The differences between these approaches are the columns, the chromatographic conditions and the MS/MS settings.

Discussion

Lines 416-419. This sentence was confusing to me “in spite” of what? I was not sure what the first part of sentence was.

Response. We slightly change the writing to show that it is unexpected that the best chromatographic conditions were obtained in the reversed system for an apparently highly hydrophilic sec

---

## [Decision Letter · Decision Letter 1]

25 Nov 2025

Dear Dr. Carazzone,

Thank you for submitting your manuscript to PLOS ONE. After careful consideration, we feel that it has merit but does not fully meet PLOS ONE’s publication criteria as it currently stands. Therefore, we invite you to submit a revised version of the manuscript that addresses the points raised during the review process.

**ACADEMIC EDITOR:** 

One of the referees did not respond to my invitation to review this version of the manuscript so I invited a third specialist to evaluate your work. While reviewer 2 was very positive towards your study, reviewer 3 raised some important concerns regarding the validation of the methods that should be addressed in the next version. Please see the comments bellow.

We look forward to receiving your revised manuscript.

Kind regards,

Renato Filogonio

Academic Editor

PLOS ONE

Journal Requirements:

Reviewers' comments:

Reviewer's Responses to Questions

**Comments to the Author**

Reviewer #2: All comments have been addressed

Reviewer #3: (No Response)

2. Is the manuscript technically sound, and do the data support the conclusions?

Reviewer #2: Yes

Reviewer #3: Yes

3. Has the statistical analysis been performed appropriately and rigorously?

Reviewer #2: Yes

Reviewer #3: Yes

4. Have the authors made all data underlying the findings in their manuscript fully available?

Reviewer #2: Yes

Reviewer #3: Yes

5. Is the manuscript presented in an intelligible fashion and written in standard English?

Reviewer #2: Yes

Reviewer #3: Yes

Reviewer #2: (No Response)

Reviewer #3: This study investigates tetrodotoxin and other alkaloids in order to identify the paralysis-producing substances present in Colostethus imbricolus. Although the toxin itself was not identified, the detection of TTX and amphibian alkaloids, as well as the chemical profiling of Colostethus imbricolus and the metabolomic comparison between males and females, provide valuable information. However, the following concerns need to be addressed.

First, the statement that “TTX was not detected” simply means that its concentration may have been below the detection limit. The detection limit of TTX under the analytical conditions used in this study must be clearly stated. In addition, it is necessary to spike the frog extract with TTX and confirm that it can be detected without ion suppression. This is especially important because the SB-CN column is being applied for the first time in this analysis.

Abstract: “A notable additional outcome of this study is the first successful separation of TTX on an SB-CN column using a normal-phase gradient, enabling a novel method for TTX-targeted separation.”

To claim novelty and usefulness of the SC-CN column for TTX-targeted analysis, validation of the method—such as detection limits, quantification capability, and reproducibility—is essential. Any statements regarding the novelty or utility of the analytical conditions should be made with caution unless supported by proper validation data.

Table 1: The functional groups (R1–R4) should be clearly defined.

Table 2: 1-Hydroxy-5,11-dideoxyTTX was structurally revised in a 2020 Journal of Natural Products paper on TTX in newts, and the previously reported structure is no longer considered valid. The corrected compound name should be used.

**Do you want your identity to be public for this peer review?** For information about this choice, including consent withdrawal, please see our For information about this choice, including consent withdrawal, please see our Privacy Policy .

Reviewer #2: No

Reviewer #3: No

---

## [Author Response · Author response to Decision Letter 2]

10 Dec 2025

Review Comments to the Author

Reviewer #2: (No Response)

Reviewer #3: This study investigates tetrodotoxin and other alkaloids in order to identify the paralysis-producing substances present in Colostethus imbricolus. Although the toxin itself was not identified, the detection of TTX and amphibian alkaloids, as well as the chemical profiling of Colostethus imbricolus and the metabolomic comparison between males and females, provide valuable information. However, the following concerns need to be addressed.

First, the statement that “TTX was not detected” simply means that its concentration may have been below the detection limit. The detection limit of TTX under the analytical conditions used in this study must be clearly stated. In addition, it is necessary to spike the frog extract with TTX and confirm that it can be detected without ion suppression. This is especially important because the SB-CN column is being applied for the first time in this analysis.

Response. Unfortunately we did not make a method validation for the analysis of TTX in either of the chromatographic methods presented, as the original aim of the study was to separate and identify the metabolites contained in the skin of C. imbricolus. Acknowledging this big limitation of our research we intentionally used the term “not detected”. Additionally, in the discussion section we openly acknowledged this important limitation of our study stating: “Unfortunately, the lack of method validation for TTX analysis, as well as the absence of proper detection and quantification limit estimations for the SB-CN system, makes it difficult to determine whether TTX is truly absent in this species or simply undetectable. However, detecting ions with nominal masses that correspond to TTX-analogues (Table 2) but with fragmentation patterns that do not match previously published data [35–38], supports the idea that C. imbricolus lacks these compounds.

Abstract: “A notable additional outcome of this study is the first successful separation of TTX on an SB-CN column using a normal-phase gradient, enabling a novel method for TTX-targeted separation.”

To claim novelty and usefulness of the SC-CN column for TTX-targeted analysis, validation of the method—such as detection limits, quantification capability, and reproducibility—is essential. Any statements regarding the novelty or utility of the analytical conditions should be made with caution unless supported by proper validation data.

Response. We really appreciate that you have noticed that this statement is not fully supported for the lack of method validation. We have edited the text to “A notable additional outcome of this study is the first successful separation of TTX on an SB-CN column using a normal-phase gradient, suggesting a potential useful approach for TTX-targeted separation.”

Table 1: The functional groups (R1–R4) should be clearly defined.

Response. We are very sorry for this mistake that was originated from a previous version of the manuscript. We have edited the figure to show exclusively the structure of TTX without R1-R4 functional groups.

Table 2: 1-Hydroxy-5,11-dideoxyTTX was structurally revised in a 2020 Journal of Natural Products paper on TTX in newts, and the previously reported structure is no longer considered valid. The corrected compound name should be used.

Response. We really appreciate that you have noticed that the name was revised. We have edited the name in Table 2 to “1-hydroxy-8-epi-5,11-dideoxyTTX”. We also noticed a typo in the structure 11-norTTX-6(R)-ol and we have corrected this in the same table.

---

## [Decision Letter · Decision Letter 2]

21 Dec 2025

Lack of tetrodotoxin analogues and individual metabolomic profiling of the cryptic frog Colostethus imbricolus

PONE-D-25-27319R2

Dear Dr. Carazzone,

We’re pleased to inform you that your manuscript has been judged scientifically suitable for publication and will be formally accepted for publication once it meets all outstanding technical requirements.

Kind regards,

Renato Filogonio

Academic Editor

PLOS One

Additional Editor Comments (optional):

Congratulations on this accepted manuscript. For future reference, I recommend taking into account referee #3 suggestion of validating the TTX analytical method (see below), which was a similar concern as referee #1. With no further comments, I hope that this acceptance will make your celebrations all the better, and I wish the authors happy holidays!

Reviewers' comments:

Reviewer's Responses to Questions

**Comments to the Author**

Reviewer #3: (No Response)

2. Is the manuscript technically sound, and do the data support the conclusions?

Reviewer #3: Yes

3. Has the statistical analysis been performed appropriately and rigorously?

Reviewer #3: N/A

4. Have the authors made all data underlying the findings in their manuscript fully available?

Reviewer #3: Yes

5. Is the manuscript presented in an intelligible fashion and written in standard English?

Reviewer #3: Yes

Reviewer #3: The reviewer considers validation of the tetrodotoxin analytical method, including determination of the detection limit, to be critically important. However, the authors have clearly stated that such verification was not performed in this study. It has also been confirmed that the description regarding the novelty of the analytical method has been revised accordingly.

**Do you want your identity to be public for this peer review?** For information about this choice, including consent withdrawal, please see our For information about this choice, including consent withdrawal, please see our Privacy Policy .

Reviewer #3: No

---

## [Editor Report · Acceptance letter]

PONE-D-25-27319R2

PLOS One

Dear Dr. Carazzone,

I'm pleased to inform you that your manuscript has been deemed suitable for publication in PLOS One. Congratulations! Your manuscript is now being handed over to our production team.

Kind regards,

on behalf of

Dr. Renato Filogonio

Academic Editor

PLOS One